# Single missense mutations in Vi capsule synthesis genes confer hypervirulence to *Salmonella* Typhi

Gi Young Lee [1] & Jeongmin Song [1]✉

Many bacterial pathogens, including the human exclusive pathogen *Salmonella* Typhi, express capsular polysaccharides as a crucial virulence factor. Here, through *S.* Typhi whole genome sequence analyses and functional studies, we found a list of single point mutations that make *S.* Typhi hypervirulent. We discovered a single point mutation in the Vi biosynthesis enzymes that control Vi polymerization or acetylation is enough to result in different capsule variants of *S.* Typhi. All variant strains are pathogenic, but the hyper Vi capsule variants are particularly hypervirulent, as demonstrated by the high morbidity and mortality rates observed in infected mice. The hypo Vi capsule variants have primarily been identified in Africa, whereas the hyper Vi capsule variants are distributed worldwide. Collectively, these studies increase awareness about the existence of different capsule variants of *S.* Typhi, establish a solid foundation for numerous future studies on *S.* Typhi capsule variants, and offer valuable insights into strategies to combat capsulated bacteria.

Many bacterial pathogens, including the human-exclusive pathogen *Salmonella enterica* serovar Typhi (*S.* Typhi), have the capsular polysaccharides (CPS) biosynthesis system as a crucial virulence factor[1–8]. Given their role in virulence and the surface display on the cognate pathogen, targeting CPS (e.g., vaccine) is a proven strategy used to prevent some of those capsulated pathogens[9–11]. The Vi CPS biosynthesis system of *S.* Typhi is encoded by 9 genes (*tviB-E* and *vexA-E*) located in the *viaB* locus within *Salmonella* Pathogenicity Island-7 (SPI-7) on the chromosome[8,12–14]. The expression of the *tviB-vexE* operon is regulated by TviA interacting with RcsB or OmpR[15–17]. TviB and TviC are responsible for producing an activated donor substrate for Vi synthesis (UDP-*N*-acetyl-D-galactosaminuronic acid), which undergoes subsequent polymerization and *O*-acetylation by TviE and TviD, respectively[8,18,19]. The role of VexE is to acylate the reducing terminal glycan of Vi, which possibly occurs before Vi polymerization, although the exact order of the Vi synthesis process remains to be characterized[20]. The synthesized Vi is transported to the outer membrane for surface display through the ATP-binding cassette (ABC) transporter composed of VexA-D[8].

*S.* Typhi causes the life-threatening systemic disease typhoid fever. Typhoid fever is a major global health concern, as evidenced by outbreaks in Southeast Asia and sub-Saharan Africa[21–24]. The World Health Organization estimates that the disease claims ~200,000 deaths per year, mostly children. *S.* Typhi is acquired through ingestion of contaminated food and water, which invades the intestinal mucosa and then spreads systemically to the liver, spleen, bone marrow, and gallbladder[25]. Following recovery, over 10% of acute typhoid cases become temporary carriers; further, a significant proportion of the individuals infected (2–6%) establish an asymptomatic chronic carriage state (colonization in the gallbladder), during which they excrete *S.* Typhi for months, and in some cases, years[26–28]. Because *S.* Typhi is a human-specific pathogen, the capacity to generate an asymptomatic carrier state is one of the critical pathogenic mechanisms leading to its persistent presence. Prolonged or persistent infection of *S.* Typhi in the gallbladder and macrophages is known to be a key feature among these persistent asymptomatic carriers[28–33]. The dynamic infectious cycle of *S.* Typhi is coordinated by many virulence factors, including Vi CPS, flagella, Type III secretion system (T3SS)

[1]Department of Microbiology and Immunology, Cornell University College of Veterinary Medicine, Ithaca, NY 14853, USA.
✉e-mail: jeongmin.song@cornell.edu

*Salmonella* Pathogenicity Island (SPI)−1 and SPI-2 effector toxins, and typhoid toxin[16,29,34–41].

During the infectious cycle of *S*. Typhi, the Vi capsule serves as a protective barrier against the host's innate immune responses, exemplified by its roles in inhibiting complement deposition and neutrophil-mediated phagocytosis[42–44]. As a result, it has been observed that acapsular (Vi-negative) *S*. Typhi is generally regarded as avirulent or less virulent in comparison to the wild-type (WT, Vi-positive) *S*. Typhi strains[42–45]. Nevertheless, there exists a notable deficiency in knowledge regarding the potential emergence of Vi capsule variants and the presence of hypervirulent variants within this category. To address this substantial disparity in knowledge, we conducted bioinformatic analyses, followed by structural and functional investigations. Our findings present experimental evidence that indicates the emergence and circulation of *S*. Typhi capsule variants including hypervirulent Vi variants.

## Results

### *S*. Typhi genes encoding the Vi biosynthesis system are hotspots for clinical missense mutations

The Vi capsule of *S*. Typhi is a crucial virulence factor that distinguishes it from other non-typhoidal *Salmonella* strains. However, despite the significant impact of *S*. Typhi infection on many millions of lives every year, whether any Vi capsule variants have emerged, aside from acapsular strains, remains completely unknown. To fill this fundamental knowledge gap, we built a bioinformatic pipeline established for the whole genome sequence (WGS) comparison analyses of 5379 *S*. Typhi clinical isolates (Fig. 1a and Supplementary Data 1). Using this tool, we conducted a re-annotation of all 5749 WGSs (available through the NCBI's GenBank database) using the Prokaryotic Genome Annotation Pipeline (PGAP) to synchronize all annotations under the same criteria. We used RefSeq's gene numbers to identify the gene pool from each strain and allocated single digit code to each gene: −1, absence of gene, 0, predicted pseudogene, 1, presence of gene (Fig. 1a and Supplementary Data 1-2). To ensure that the dataset includes *S*. Typhi WGSs only, we included two additional computational filters in the pipeline: (i) SeqSero2[46] and (ii) typhoid toxin subunits (*cdtB, pltA,* and *pltB*)[47] to filter out non-Typhi strains, resulting in a total of 5,379 strains that were used in this study (Fig. 1a and Supplementary Data 1).

To gain a comprehensive understanding of whether there are mutations in Vi synthesis and its related genes, we analyzed missense point mutations that occurred in the genes for Vi regulation (*rcsB, rcsC, envZ, ompR,* and *tviA*), synthesis (*tviB, tviC,* and *tviE*), modification (*tviD* and *vexE*), and transport (*vexA, vexB, vexC,* and *vexD*)[8,12,13,18,48,49] (Fig. 1b). The resulting clinical missense mutation frequency map reveals that nearly all mutations occurred in two genes encoding TviE and TviD (Fig. 1c and Supplementary Data 3).

Contrary to non-synonymous mutations (Fig. 1c), synonymous mutations in the *tviE* gene (1.3%, 10 out of 747 SNPs) and the *tviD* gene (13.7%, 177 out of 1289 SNPs) are much less frequent than non-synonymous mutations (Fig. 1d). In addition, certain strains possess the Vi biosynthesis genes as pseudogenes due to the presence of internal stop and frameshift mutations (Supplementary Data 2). As reported previously[50,51], our WGS dataset includes clinical isolates that lack the entire Vi synthesis operon (Supplementary Data 2). *S*. Typhi mutants containing synonymous mutations and predicted pseudogenes are expected to exhibit phenotypic similarities to WT and acapsular *S*. Typhi strains, respectively. Consequently, these strains were excluded from further characterization in this study.

Given their predicted roles within *S*. Typhi in polymerizing Vi and/or adding the *O*-acetyl (*O*Ac) moiety at the C3 position[19], we reasoned that *S*. Typhi variants carrying a single amino acid point mutation in TviE or TviD express a variant form of Vi on the surface of *S*. Typhi. To test the hypothesis, we chose the top 21 clinical missense mutations that are most frequently found in *tviE* and *tviD*, as well as 1 *tviA* mutant.

We have generated these 22 mutants of *S*. Typhi, each carrying a single point mutation in *tviE, tviD,* or *tviA*. These mutations were introduced into the isogenic background of the strain ISP2825, which is a clinical isolate of *S*. Typhi[52] (Fig. 1d lower panel and Supplementary Table 1).

We first examined Vi production in various salt concentrations (a known environmental cue altering *S*. Typhi Vi expression)[53]. We observed a higher Vi expression and 3-*O*-acetylation in *S*. Typhi cultured in a medium containing 60–150 mM NaCl with a peak expression at 90 mM NaCl, while a decreased Vi expression was seen in *S*. Typhi cultured in salt-free and high salt-containing media (Fig. 1e, f, and Supplementary Figs. 1 and 2). Based on this result, we used 86 mM NaCl LB (a.k.a., low salt LB or Lennox LB) throughout this study, unless specified. Using the validated antibodies specifically detecting pan-Vi (both unmodified and *O*Ac-Vi) or *O*Ac-Vi (Supplementary Fig. 1), we assessed 22 *S*. Typhi mutants and found that, except *tviA* V62I, each of these single point mutations leads to the variant form of Vi, whose intensity, length, and/or acetylation are different from WT Vi (Fig. 1g, k and Supplementary Figs. 2–5). Unlike other mutations, we found that *tviA* V62I is not a phenotype-changing mutation (Supplementary Fig. 3). Based on the obtained results, the 21 mutations were categorized into three groups: hypo, hyper, and intermediate Vi capsule variants (Fig. 1g–k and Supplementary Figs. 2, 4, and 5). These groups correspond to mutations that result in less than WT (hypo Vi, pink), more than WT (hyper Vi, lilac blue), and slightly more than WT (intermediate Vi, light gray) in terms of Vi intensity, length, and/or acetylation changes (Fig. 1g–k and Supplementary Figs. 2, 4, and 5).

To conduct a further assessment of the variant form of Vi, we employed alcian blue 8GX. This particular stain is widely used for detecting acidic, neutral, sulfated, and phosphate polysaccharides as well as glycosaminoglycans[54]. One-half of the cultures of WT and mutant *S*. Typhi strains were analyzed by alcian blue SDS-PAGE, while the remaining half of the samples were analyzed using immunoblots simultaneously. Their migration differences were similar in both results (Supplementary Fig. 6). Due to numerous non-specific background signals present in alcian blue-stained SDS-PAGE, in contrast to the specific detection of Vi observed in immunoblot results, we used Vi-specific antibody-mediated methods in this work (Supplementary Fig. 6). We further found that the observed modifications in Vi CPS resulting from individual point mutations are unlikely to be attributed to the altered expressions of TviD or TviE, as we detected comparable mRNA expressions of *tviD* and *tviE* between WT and capsule variant *S*. Typhi strains (Supplementary Fig. 7). Furthermore, given that other bacteria's hypercapsule variants, such as *Klebsiella pneumoniae* and *Acinetobacter baumannii*, are correlated with a mucoidy characteristic, we assessed the mucoidy of hyper Vi capsule variants. We found that, unlike other bacteria (e.g., *K. pneumoniae* and *A. baumannii*), the hyper Vi capsule variation in *Salmonella* strains is not associated with mucoidy (Supplementary Table 2).

### TviE function is altered by clinical missense mutations

To determine the underlying molecular mechanism of how a single amino acid sequence variation leads to the variant form of Vi, as well as to identify a plausible molecular evolution strategy used by *S*. Typhi, we predicted the 3-dimensional (D) structure of TviE by running two machine-learning-based structure prediction programs AlphaFold v2 and RosettaFold[55,56] (Fig. 2a). We found a strong consensus between the two resulting structures, particularly for the C-terminal and internal regions of TviE, where those clinical missense mutations are positioned (Fig. 2b, c). When we positioned all TviE single point mutations on the 3D structure (which we note that each clinical isolate of *S*. Typhi carries one single point mutation), we noticed that those mutations are not only positioned on the one-themed location, referred to as the "horizontal groove", but also result in a drastic change of their electric surface charge distribution of the horizontal groove (Fig. 2b, c). Based on these observations, it can be deduced that amino acids in the

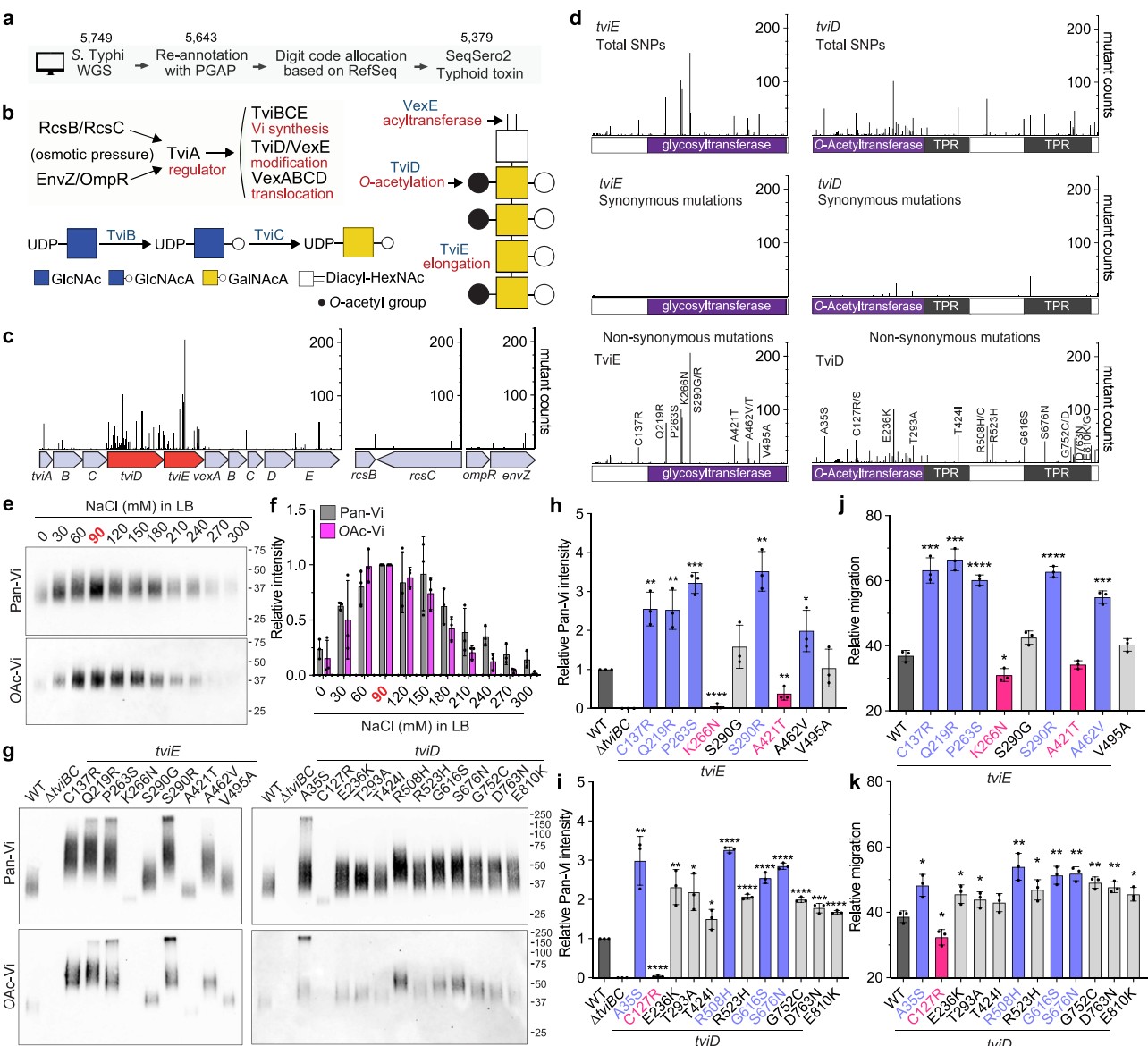

**Fig. 1 | S. Typhi genes encoding the Vi biosynthesis system are hotspots for clinical missense mutations. a** A schematic workflow of the bioinformatic pipeline used to analyze S. Typhi WGSs. **b** A working model for S. Typhi Vi gene regulation, synthesis, modification, and transport. Note that the exact order of the Vi synthesis process remains to be characterized. GlcNAc, N-acetyl-glucosamine. GlcNAcA, N-acetyl-glucosaminuronic acid. GalNAcA, N-acetyl-galactosaminuronic acid. Diacyl-HexNAc, diacyl-N-acetyl-hexosamine. **c**, Clinical missense mutation frequency map that we have generated in this study. Note that the peaks are based on protein-level mutations. **d** Total SNPs (upper panel), synonymous (middle panel), and non-synonymous (lower panel) mutations found in the *tviE* and *tviD* genes. Note that the peaks for total SNPs and synonymous SNPs are based on nucleotide-level mutations, whereas the peaks for non-synonymous SNPs are based on protein-level mutations. Y-axis, numbers of S. Typhi clinical isolates carrying the indicated single point mutation of the indicated genes. X-axis, the position of the single point mutations. TPR, tetratricopeptide repeat. **e** Immunoblots assessing Vi produced by WT S. Typhi cultured in LB containing indicated NaCl concentrations. **f** Quantification results of three independent experiments associated with **e**. **g** Immunoblots assessing Vi produced by S. Typhi WT, Δ*tviBC*, and clinical missense mutants cultured in LB containing 86 mM NaCl. **h** Vi intensity quantification results of three independent experiments associated with **g** left panel. **i** Vi intensity quantification results of three independent experiments associated with **g** right panel. **j** Vi migration/length quantification results of three independent experiments associated with **g** left panel. **k** Vi migration/length quantification results of three independent experiments associated with **g** right panel. Bars in graphs **f** and **h**–**k** represent the mean ± standard deviation (SD). Two-tailed student *t* tests between WT and indicated mutants were performed. *$P < 0.05$; **$P < 0.01$; ***$P < 0.001$; ****$P < 0.0001$. The positions of the protein ladders in **e** and **g** are indicated for guidance regarding relative glycan migration differences. See also Supplementary Figs. 1–7 and Supplementary Tables 1, 2, and Data 1–3. Source data are provided as a Source Data file.

horizontal groove likely play an important role in the function of the TviE enzyme, as this horizontal groove is situated after the catalytic residues (E483 and E491)[19]. In this possible scenario, the presence of a single amino acid variation in the groove may be able to alter Vi elongation (Fig. 2c, d). Our finding in this study indicate that Vi products produced by WT *tviE* K266, *tviE* K266N (hypo Vi capsule clinical isolate), *tviE* K266R (positively charged basic a.a.), and *tviE* K266E (acidic a.a.) are in agreement with the proposed TviE acceptor binding

mode (Fig. 2e and Supplementary Fig. 8a). This finding supports our hypothesized bacterial evolution mechanism, indicating the impact of specific amino acid residues on the elongation process of Vi. Nevertheless, further research is necessary to formally test this potential mechanism and other alternative mechanisms.

To gain an understanding of the mechanism of mutations that confer hypo and hyper Vi capsule phenotypes, we generated the in vivo molecular tool using the whole S. Typhi bacteria. We first made a

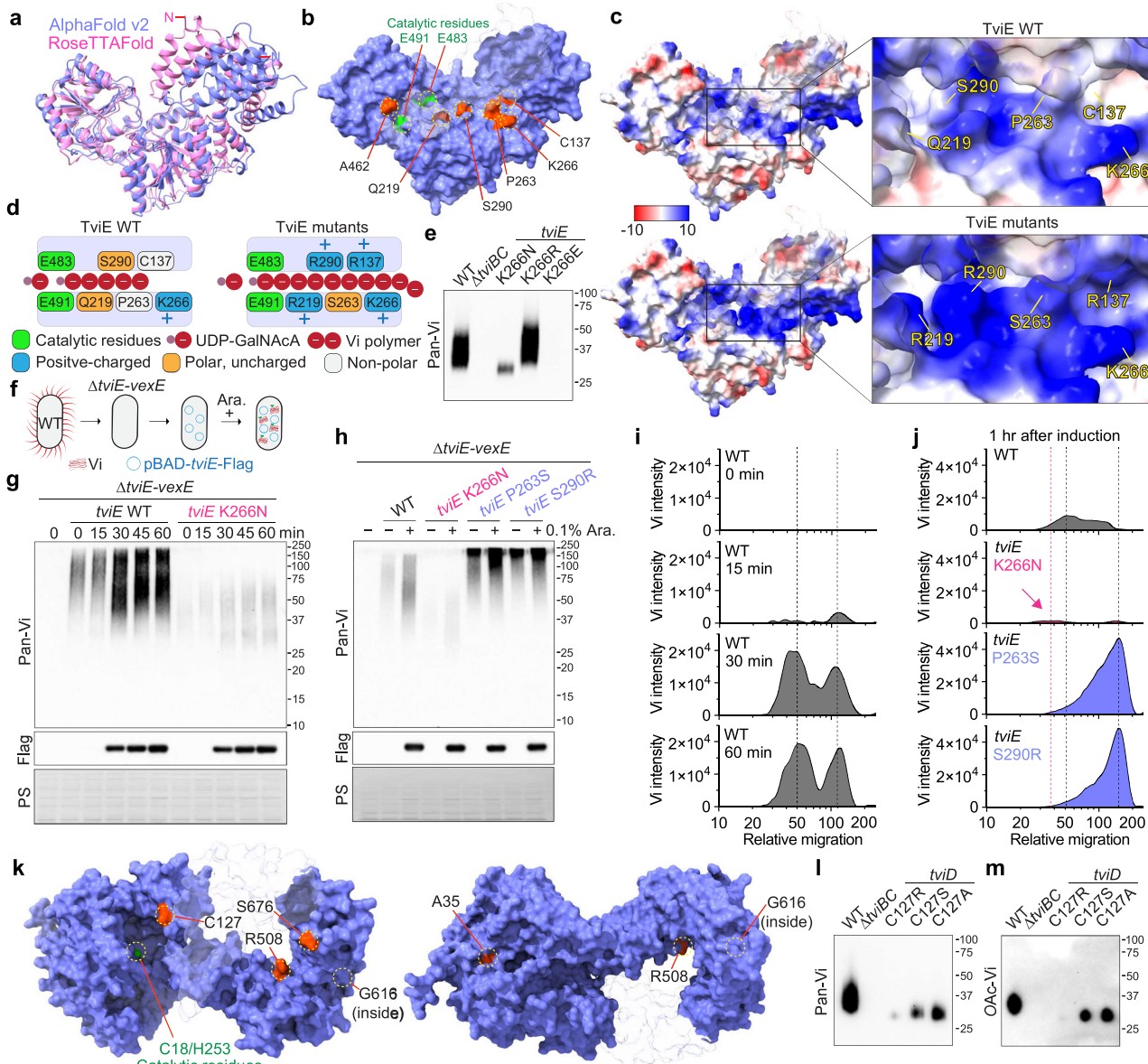

**Fig. 2 | The presence of clinical missense mutations leads to alterations in the mechanism of action of TviE and TviD. a** Overlay ribbon diagrams of TviE structures. Pink, TviE structure predicted by RoseTTAFold. Lilac blue, TviE structure predicted by AlphaFold v2. N, N-terminus of TviE. **b** Clinical missense mutations on the predicted TviE structure (red residues). E491 and E483 are catalytic residues. Lilac blue, the molecular surface of AlphaFold-predicted TviE structure. Light gray/semi-transparent, M1-M45, T214-R218, and R406. **c** Surface charge distributions of the TviE WT and mutants. Note that R137, R219, R290, and S263 are indicated in the same structure in this figure. Blue, positive charge. Red, negative charge. Semi-transparent, M1-M45, T214-R218, and R406. **d** A schematic cartoon depicting the predicted Vi extension process occurring on the horizontal groove of TviE WT and mutants. **e** Immunoblot assessing the role of TviE Lys residue at 266 in Vi polymerization control. *S.* Typhi WT and indicated *tviE* mutants were generated and characterized. **f** A schematic cartoon illustrating the in vivo molecular tool

developed to evaluate TviE mutants. **g, h** Time-course immunoblots of WT *tviE* and *tviE* mutants. PS, Ponceau S stained membranes which are to demonstrate comparable sample loading. Arabinose (0.1%) was used. **i** Time-course histograms of Vi length (X-axis) and amount (Y-axis) produced by *S.* Typhi Δ*tviE-vexE* carrying pBAD-*tviE*-Flag. Relative migration values are correlates of the apparent sizes of protein standards in KDa. **j** Histograms of Vi length (X-axis) and amount (Y-axis) produced by *S.* Typhi Δ*tviE-vexE* carrying pBAD-*tviE*-Flag or pBAD-*tviE* K266N, P263S, or S290R-Flag. One hour after 0.1% arabinose induction. **k** Clinical missense mutations on the predicted TviD structure (red residues). C18 and H253 are catalytic residues. Semi-transparent, the residues A701-S831. **l, m** Immunoblots assessing the role of TviD Cys residue at 127 in Vi length and *O*-acetylation. Three independent experiments were performed for **e, g, l,** and **m**. The positions of the protein ladders in **e, g, h, l,** and **m** are indicated for guidance regarding relative glycan migration differences. Source data are provided as a Source Data file.

clean-deletion of 6 consecutively located genes within the *viaB* locus in *S.* Typhi, *tviE, vexA, vexB, vexC, vexD,* and *vexE,* designated as *S.* Typhi Δ*tviE-vexE* (Fig. 2f). The generated hexaduple mutant was transformed with a plasmid expressing pBAD-*tviE*-Flag (Fig. 2f). pBAD promoter was used to fine-tune the expression of TviE, while the Flag tag was added to monitor the expression of TviE upon arabinose treatment. Due to the lack of the transporter of the synthesized Vi in this engineered strain (VexA-D), the synthesized Vi is retained within *S.*

Typhi, enabling a time-course evaluation of the synthesized Vi (Fig. 2f). Using this tool developed, we assessed the impact of 3 representative mutants of TviE, K266N (hypo Vi), P263S (hyper Vi), and S290R (hyper Vi), on Vi synthesis. We found that Lys at position 266 is critical for the mode of action for TviE, as K266N resulted in hypo Vi product with a reduced expression (Fig. 2g and Supplementary Fig. 8b). In contrast, we found that the mutations associated with the hyper Vi capsule product (P263S and S290R) resulted in a stronger expression of hyper

Vi than WT *S.* Typhi (Fig. 2h–j and Supplementary Fig. 8c), indicating that Pro at position 263 and Ser at position 290 are also critical for TviE function.

## Clinical missense mutations of TviD are distributed throughout its 3D structure, yet it seems to evolve to acquire specific amino acid residues for specific variants

To gain insights into the effects of Tvi D's clinical missense mutations on Vi production, we have employed the predicted structure of TviD to superimpose the frequently observed clinical missense mutations (Fig. 2k). In contrast to the frequent clinical missense mutations observed in TviE, the mutations in TviD are distributed throughout its 3D structure, specifically on its surface except G616. The surface and distributed location of these mutations suggest that multiple evolutionary events have contributed to their emergence. These mutations are likely to have impacts on various aspects of Vi synthesis process, including TviD's catalytic activity and the formation of the predicted multimeric, multiprotein enzyme complexes for Vi production. Further investigation is required to comprehend the specific effects of each mutant on Vi production. However, for the purpose of this study, we have chosen to focus on the C127R variant (hypo Vi capsule). This particular variant is situated in close proximity to the catalytic site of TviD, approximately 17 Å and 21 Å away from H253 and C18, respectively (Fig. 2k). Our aim was to experimentally assess the influence of specific amino acid residues, as observed in clinical isolates, on Vi production (Fig. 1g). We found that the clinical missense mutation C127R significantly impairs the function of TviD (Fig. 2l, m and Supplementary Fig. 8d, e). In contrast, experimental mutations, specifically C127S and C127A, result in a modest decrease (Fig. 2l, m and Supplementary Fig. 8d). These findings support bacterial evolution, as the substitution of residue C127 with residue R appears to be one of the best residues to drastically change the phenotype at this position. Based on the close distance to the catalytic residues, one possible mechanism is that the C127 residue could be involved in facilitating the transfer of the acetyl group to Vi during the acetylation process of TviD. To test this prediction and potential alternative mechanisms, more research is required.

## *S.* Typhi hypo Vi capsule variants possess several critical pathogenic characteristics, including serum resistance and increased invasion

We have observed similar growth patterns in both the *S.* Typhi WT and Vi variant strains (Fig. 3a). This suggests that there is no discernible growth defect associated with the production of variant forms of the Vi capsule in axenic cultures. It is equally important to ascertain whether Vi variants, similar to WT *S.* Typhi, possess pathogenic properties. We reasoned that these hypo and hyper Vi capsule variants are pathogenic since these clinical mutations are originated from human infection sites (Fig. 3b and Supplementary Data 1). In particular, in the so-called "hypo Vi capsule" group (>3% of 5379 clinical isolates; Fig. 3b), *S.* Typhi hypo Vi variants exemplified by *tviE* K266N and *tviD* C127R have shorter Vi, which, therefore, we hypothesized that the hypo Vi capsule variants are able to avoid host innate immune defense mechanisms like WT (e.g., serum resistance), while these variants invade host cells better than WT. In the "hyper Vi and intermediate Vi capsule" group (>16%; Fig. 3b), S. Typhi hyper Vi variants (e.g., *tviE* C137R, *tviE* Q219R, *tviE* P263S, and *tviE* S290R) are expected to demonstrate an increased capacity to evade neutrophil-mediated phagocytosis and other immune responses. We predict that this is attributed to the presence of long Vi and shed Vi. The "intermediate Vi capsule" group is anticipated to have an intermediate phenotype between WT and hyper Vi capsule variants.

To demonstrate hypo Vi capsule variants, we conducted image cytometry using antibodies specific to Vi and LPS O9 surface glycans (Fig. 3c, d). We indeed observed that the hypo Vi capsule variants *tviE*

K266N and *tviD* C127R of *S.* Typhi exhibit minimal Vi expression on the bacterial surface (Fig. 3c, d). The signal intensity of Vi is significantly lower than that of the WT bacteria, but differs from the complete absence of Vi observed in the acapsular *S.* Typhi *ΔtviBC* strain (Fig. 3c, d).

We conducted an investigation to assess the serum resistance of the hypo Vi capsule variants. This was accomplished by utilizing human sera obtained from unvaccinated healthy individuals who had not been immunized against typhoid fever. In contrast to the acapsular *S.* Typhi *ΔtviBC* strain, the survival of *S.* Typhi strains carrying *tviD* C127R and *tviE* K266N, which are two hypo Vi capsule variants, remained robust even after 2 h of incubation at 37 °C in the presence of 90% healthy human sera. These strains exhibited only a minor decrease in survival when compared to the WT *S.* Typhi strain (Fig. 3e, f). The specificity of the findings (Fig. 3e, f) is supported by the use of PBS-only or heat-inactive human sera (iHS) samples. The findings suggest that hypo Vi capsule is effective in conferring resistance to serum, thereby distinguishing the hypo Vi capsule variants from the acapsular counterparts. Comparable results of serum resistance of the hypo Vi capsule variants were also observed with human sera from vaccinated healthy individuals who had been immunized against typhoid fever (Fig. 3g, h).

One significant pathogenic characteristic that sets *S.* Typhi apart from other bacterial pathogens with capsules is its superior ability to invade host cells. Our research, conducted using quantitative microscopy-based infection assays, revealed that hypo Vi capsule variants significantly enhanced the infectivity of *S.* Typhi by 2-3 fold (Fig. 3i, j). This suggests that *S.* Typhi Vi variants with hypo Vi exhibit an expanded pathogenic characteristic. By employing quantitative inside-outside staining, we have additionally discovered that hypo Vi capsule variants of *S.* Typhi exhibit a higher degree of adherence to host cells compared to the WT strain (Fig. 3k, l), which is consistent with their notable increase in invasion. These results are in agreement with the findings from bacterial colony forming unit (CFU) assays (Fig. 3m). Next, we proceeded to assess whether the hypo Vi capsule variant demonstrates an increased infection using Cmah null mice. The Cmah null mice are immunocompetent while exclusively expressing glycans that are terminated in N-acetylneuraminic acid (Neu5Ac) due to the lack of the CMP-Neu5Ac hydroxylase (Cmah), which closely resemble those observed in humans[57]. In contrast, the C57BL/6 mice (having the functional Cmah enzyme) exhibit glycans that are terminated in both Neu5AC and N-glycolylneuraminic acid (Neu5Gc). It is important to acknowledge that the brain of C57BL/6 mice, like that of humans and Cmah null mice, exclusively expresses Neu5Ac, highlighting that the expression of Cmah varies across different cells and tissues in mice[58]. Due to a closer glycan resemblance of Cmah null mice to humans, these animals have been used for studying the virulence of human pathogens[47,59–64]. We found that mice infected with the hypo Vi capsule variant *tviE* K266N had an increased infection than mice infected with WT (Fig. 3n); this difference is very similar to what we saw in vitro, where the infection was two to three times greater (Fig. 3j, m).

## The hyper Vi capsule variants of *S.* Typhi are hypervirulent with both bacteria-associated and shed Vi contributing to their heightened virulence

To assess the pathogenicity of the hyper Vi capsule variants in vivo, we infected Cmah null mice with WT or hyper Vi capsule (*tviE* P263S) *S.* Typhi. Mice infected intraperitoneally with $8 \times 10^5$ of the hyper Vi capsule variant (*tviE* P263S) showed severe clinical signs (including diarrhea, overall weakness, and eye discharge) and liver abscesses, resulting in the death of 50% of the infected mice by day 3 post infection (Fig. 4a–c, and Supplementary Table 3). Organ CFUs of the hyper Vi capsule variant were markedly increased, which was 100,000-fold higher than that of WT at day 3 post infection (Fig. 4d). It is notable that 100% of mice infected with the hyper Vi capsule *S.* Typhi

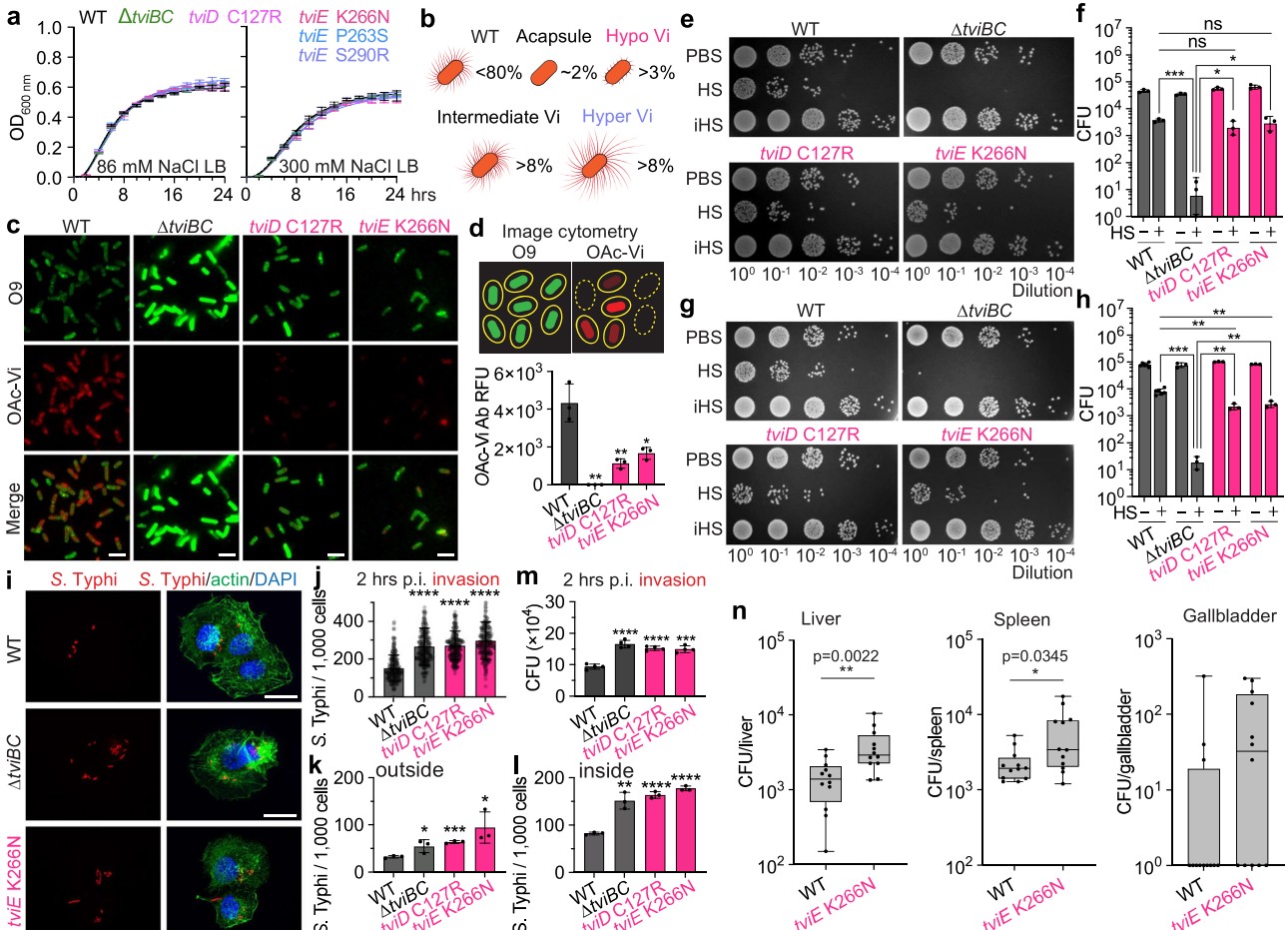

**Fig. 3 | *S*. Typhi hypo Vi capsule variants possess several critical pathogenic characteristics, including serum resistance and increased invasion. a** Growth curves of *S*. Typhi WT, Δ*tviBC*, and indicated clinical missense mutants cultured in 86 mM NaCl LB (Lennox LB) or 300 mM NaCl LB. Lines represent the mean ± SD. The results are from four independent experiments. **b** Schematic illustration depicting the capsule variants. Numbers in % indicate the % of capsule variants found among 5379 clinical isolates. **c** Representative images of image cytometry analyses on *S*. Typhi WT, Δ*tviBC*, *tviD* C127R, and *tviE* K266N. scale bars, 5 μm. **d** Quantification results of three independent experiments associated with **c**. **e**–**h** Human serum resistance assays of *S*. Typhi WT and mutants. Ten-fold dilutions of bacterial cultures, which were incubated at 37 °C for 2 h in the presence of 90% non-vaccinated (**e**, **f**) or vaccinated (**g**, **h**) human sera, plated on LB agar plates, incubated overnight, and photographed. PBS no sera, HS human sera, iHS heat-inactivated human sera. **i** Representative fluorescence microscopy images of bacterial invasion into host cells. Henle-407 cells were infected for 2 h with 15 m.o.i. of

*S*. Typhi WT, Δ*tviBC*, *tviE* P263S, or *tviE* K266N. Scale bars, 20 μm. **j** Quantification results of three independent experiments associated with **i**. 2 h after infection. **k**, **l** Quantification results of three independent experiments assessing outside (**k**) and inside (**l**) *S*. Typhi 1 h after infection. **m** Quantification results of three independent experiments assessing CFUs of intracellular *S*. Typhi 2 h after infection. Bars represent the mean ± SD except graph **f** and **h** that show the geometric mean ± SD. Two-tailed t-tests between WT and indicated strain were performed unless otherwise indicated. *$P < 0.05$; **$P < 0.01$; ***$P < 0.001$; ****$P < 0.0001$. ns not significant. **n**, CFU results that were recovered from the liver, spleen, and gall-bladder 3 days after infection ($n = 12$ mice). The box and whiskers plot shows all data points, with whiskers ranging from the lowest to the highest values. The box depicts the interquartile range (from the 25th to the 75th percentiles), while the central line indicates the median value. Two-tailed Mann–Whitney tests were performed. Source data are provided as a Source Data file.

successfully colonized in the gallbladder (Fig. 4d). Intriguingly, consistent with the observed superior gallbladder colonization of the hyper Vi capsule *S*. Typhi, a recent WGS analysis study on human gallbladder carriage *S*. Typhi isolates found other hyper Vi capsule mutations, *tviD* R508H, *tviE* C137R, and *tviE* A462V in several gall-bladder carriage isolates[33,65].

To further evaluate the impact of the hyper Vi capsule variant on in vivo infection of *S*. Typhi, we conducted co-infection studies in mice. The co-infection involved the simultaneous administration of WT and hyper Vi capsule *S*. Typhi strains at a 1:1 ratio ($4 × 10^5$ each). The total inoculum used for the co-infection was equivalent to the single infection inoculum described in Fig. 4a–d. We found that the organ CFUs of hyper Vi capsule *S*. Typhi were consistently 2–4 log higher than WT (Fig. 4e). Intriguingly, there was an observed increase in the infection of co-infected WT *S*. Typhi (Fig. 4f–h). The phenotype was not

observed when WT *S*. Typhi was co-infected with the hypo Vi capsule variant of *S*. Typhi (Fig. 4f–h), indicating that the increased infection of co-infected WT *S*. Typhi is associated with hyper Vi capsule.

Based on the findings derived from co-infection studies, it can be inferred that the excretion of Vi antigen by hyper Vi capsule bacteria potentially contribute to the heightened infection observed in co-infected WT *S*. Typhi. We conducted an assessment to determine if the hyper Vi capsule variant exhibits substantial Vi shedding. Our findings revealed that *S*. Typhi strains with hyper Vi capsule shed significantly higher quantities of the hyper Vi capsule (Fig. 4i, j and Supplementary Figs. 9 and 10). On the other hand, the shedding of Vi was found to be negligible in the case of hypo Vi capsule *S*. Typhi (Fig. 4i, j). We next investigated the impact of shed Vi on *S*. Typhi infection; we conducted a purification process on the shed Vi. This involved clearing Vi by filtration, removing the majority of LPS, and subsequently purifying it

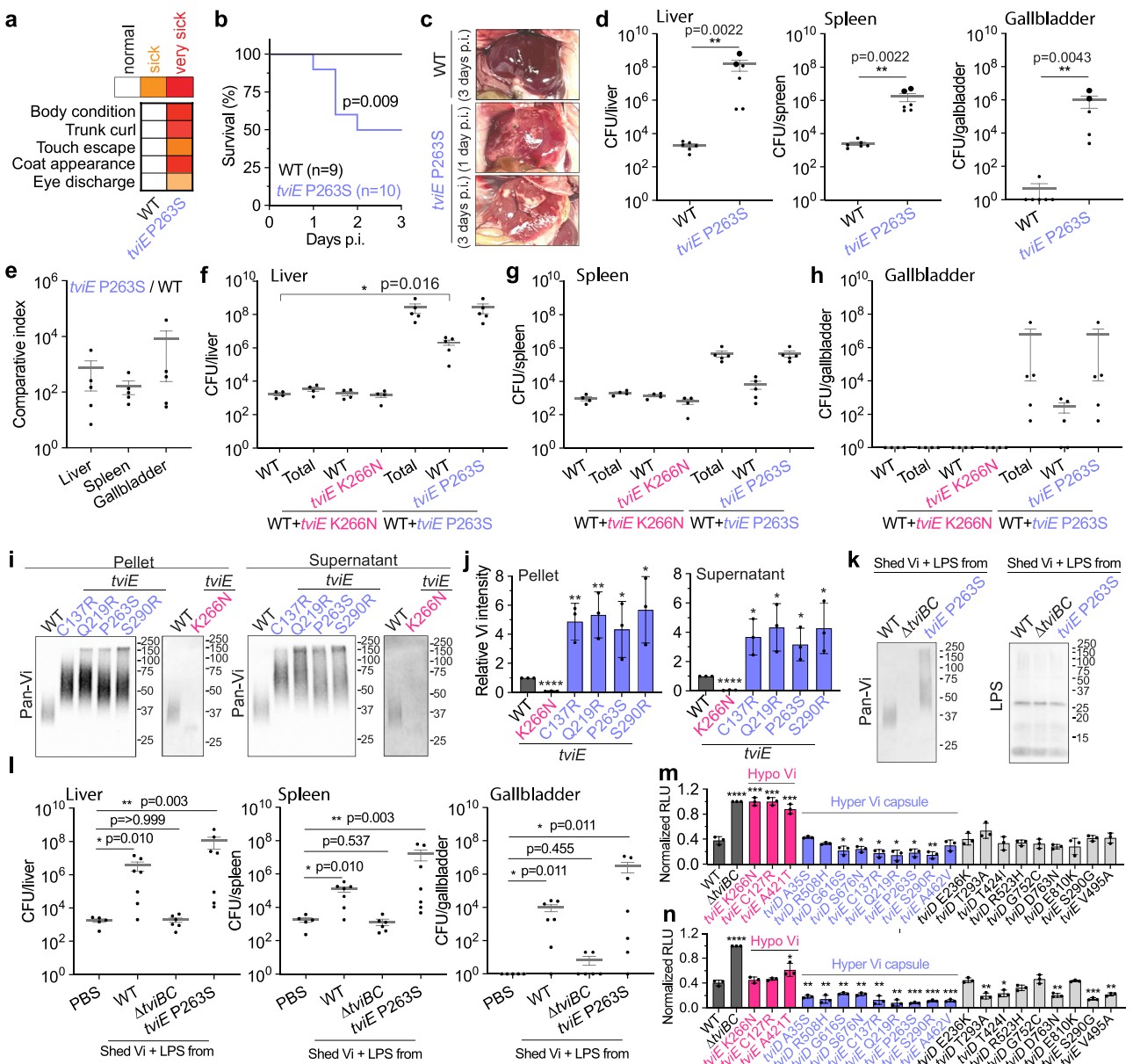

**Fig. 4 | The hyper Vi capsule variants of *S.* Typhi are hypervirulent, with both bacteria-associated and shed Vi contributing to their heightened virulence.**
**a–d** SHIRPA test (**a**), percent survival (**b**), and liver abscess (**c**) from Cmah-null mice infected intraperitoneally with 8 ×10⁵ WT (*n* = 9), or *tviE* P263S (*n* = 10). **d** CFU assays in the liver, spleen, and gallbladder 3 days after infection. Big dots represent CFU numbers from two severely sick mice that died on day 1.5. Note that the organ CFUs of WT *S.* Typhi on day 3 are the same as those of Fig. 3n WT *S.* Typhi on day 3 because these studies were carried out concurrently. **e–h** Chloramphenicol-resistant or Kanymycin-resistant gene was inserted downstream of the pseudo-nized *pagL* gene of WT (Cm^R), *tviE* K266N (Kan^R), or *tviE* P263S (Kan^R). Cmah-null mice were infected intraperitoneally with 4 ×10⁵ WT *S.* Typhi, or with a combination of WT+*tviE* K266N (4 ×10⁵ each; *n* = 4 mice), or WT+*tviE* P263S *S.* Typhi strains (4 ×10⁵ each; *n* = 5 mice). Comparative index (**e**) and CFUs in the liver (**f**), spleen (**g**), and gallbladder (**h**) 3 days after infection. **i** Immunoblots assessing the shedding of Vi by *S.* Typhi *tviE* K266N, *tviE* C137R, *tviE* Q219R, *tviE* P263S, and *tviE* S290R. **j**, Quantification results of three independent experiments associated with **i**. **k**, **l** In vivo effects of shed Vi on bacterial infection. Immunoblot analysis of Vi

preparations and residual LPS (**k**). Shed Vi (obtained from ~1.6 × 10⁹ of the indicated bacteria) was prepared in 100 µl PBS, which was administered to mice along with 8 ×10⁵ WT *S.* Typhi (*n* = 5 mice for the PBS (no shed Vi) group, *n* = 7 mice for the shed Vi from WT *S.* Typhi, *n* = 6 mice for the equivalent sample from *S.* Typhi Δ*tviBC*, and *n* = 7 mice for the shed Vi from *S.* Typhi *tviE* P253S). CFU results that were recovered from the liver, spleen, and gallbladder 24 h after infection (**l**). **m**, **n** ROS burst from neutrophils incubated with the indicated strain after opsonization with un-vaccinated (**m**) or vaccinated (**n**) human sera. Neutrophils were obtained from three blood donors. Three independent experiments were performed. RLU, relative luminescence unit. Bars/lines represent the mean ± SD (for **j**, **m**, and **n**) and the mean ± SEM (for **d–h** and **l**). Log rank tests (**b**), two-tailed t-tests (**j**) and two-tailed Mann–Whitney tests (all others) were used to compare WT to the designated strain. *P < 0.05. **P < 0.01. ***P < 0.001., ****P < 0.0001. See also Supplementary Table 3. The positions of the protein ladders in **i** and **k** are indicated for guidance regarding relative glycan migration differences. Source data are provided as a Source Data file.

using size exclusion chromatography. This Vi preparation (with low LPS contamination, Fig. 4k) was utilized for conducting infection studies in mice. The mice were co-injected via IP with this Vi preparation and WT *S.* Typhi strain (Fig. 4l). The amount of Vi preparation used was

corresponding to the surface Vi of ~1.6 × 10⁹ *S.* Typhi, which was co-injected with 8 × 10⁵ *S.* Typhi WT in 100 µl of PBS into each mouse. Vi preparations derived from both WT and hyper Vi capsule variants of *S.* Typhi were utilized. To eliminate the possibility of LPS-mediated

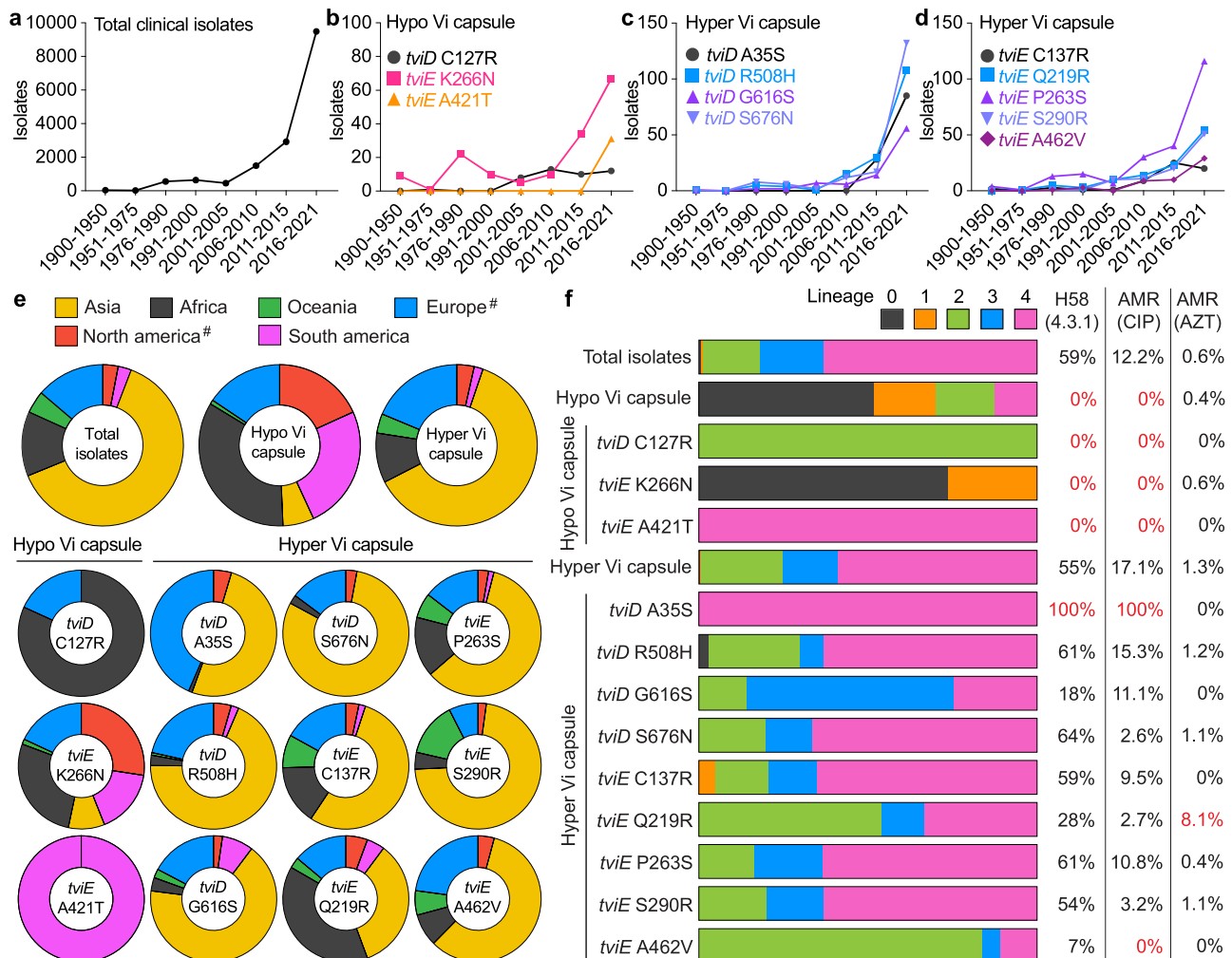

**Fig. 5 | Global distribution of hypo and hyper Vi capsule variants.** *S.* Typhi WGSs from NCBI and Pathogenwatch were merged and evaluated. Duplicate samples were eliminated, yielding a total of 16,368 strains. The years in which *S.* Typhi Vi variants have been observed. Total (**a**), hypo Vi capsule (**b**), and hyper Vi capsule (**c**, **d**) clinical isolates isolated from 1916 to 2021. **e** Global distribution of the capsular variants. #, the isolation of *S.* Typhi from North America (USA and Canada) and Europe (UK) may be related to travel. **f** Capsular variant genotypes and related antimicrobial resistance. CIP ciprofloxacin, AZT azithromycin. See also Supplementary Data 4. Source data are provided as a Source Data file.

effects, we have also included the equivalent preparation from acapsular *S.* Typhi *ΔtviBC*; as shown in Fig. 4k, comparable levels of LPS were detected across all three samples used. We observed that the administration of Vi preparations significantly increased the infection of WT *S.* Typhi when compared to the absence of additional Vi (Fig. 4l). We also observed that the Vi preparation obtained from hyper Vi capsule *S.* Typhi tends to exhibit a stronger increase effect than the Vi preparation derived from WT *S.* Typhi (Fig. 4l). The results collectively indicate that the hyper Vi capsule variant increases infectivity, and the shed Vi from the hyper Vi capsule mutant can contribute to the observed enhancement of in vivo infection.

To understand the role of bacteria-associated hyper Vi capsule, as well as to gain insights into their relevance to human infection, human peripheral blood neutrophils were isolated from healthy blood donors and assessed neutrophil reactive oxygen species (ROS) burst[36,66]. *S.* Typhi strains were opsonized with 10% human sera obtained from unvaccinated healthy individuals who had not been immunized against typhoid fever. We found that ROS bursts from neutrophils incubated with hyper Vi capsule variants were much less susceptible to phagocytosis than those from neutrophils incubated with WT, acapsule, or hypo Vi capsule variants (Fig. 4m). Intermediate Vi capsule variants had intermediate phenotypes, as some

phenocopied hyper Vi capsule variants while some phenocopied WT (Fig. 4m). Comparable results were also observed with human sera from vaccinated healthy individuals who had been immunized against typhoid fever (Fig. 4n). This study collectively illustrates that hyper Vi capsule *S.* Typhi is a hypervirulent pathogen possessing specific enhanced pathogenic traits. These characteristics may potentially benefit other Vi variants and even other pathogens in cases of co-infection.

## The hypo Vi capsule variants have primarily been identified in Africa, whereas the hyper Vi capsule variants are distributed worldwide

To gain insights into the occurrence, distribution, and persistence of hypo and hyper Vi capsule *S.* Typhi variants in human populations, along with their potential association with antibiotic resistance, we conducted comprehensive analyses on clinical isolates of *S.* Typhi. Our analysis included an initial dataset of 5379 isolates (Fig. 1a), as well as 13,081 WGSs obtained from PathogenWatch. It has been observed in this study that the hypo and hyper Vi capsule variants, which were characterized, have been persistently present in human populations throughout the entire century (Fig. 5a–d). This indicates the enduring presence of both hypo and hyper Vi capsule variants.

Based on our analyses, it has been determined that the hypo Vi capsule variants carrying *tviD* C127R and *tviE* K266N have been prevalent in various African countries for the past century (Fig. 5e and Supplementary Data 4). In contrast, the hypo Vi capsule variant carrying *tviE* A421T has emerged in 2017, which has been observed in Brazil (Supplementary Data 4). Consistent with their dissemination findings, the hypo Vi capsule variants appear to undergo convergent evolution, as the two previous variants of hypo Vi capsules are classified as *S.* Typhi lineage 0, 1, and 2, while the variant *tviE* A421T falls under lineage 4 (Fig. 5f). Our analysis did not observe any correlation between hypo Vi capsule variants and antibiotic resistance (Fig. 5f). The existence of hypo Vi capsule variants over the past century, despite their susceptibility to drugs, aligns with its increased infectivity that we observed in a mouse model.

In contrast to the limited distribution of hypo Vi capsule variants, our analysis indicates that hyper Vi capsule variants are widely distributed on a global scale (Fig. 5e and Supplementary Data 4). Hyper Vi capsule variants have also been observed persistently for the entire century, but their distribution is limited to lineages 2, 3, and 4, with the majority being found in lineage 4 (Fig. 5f). Notably, the hyper Vi capsule *tviD* A35S was identified in 2013 and has predominantly been observed in India (Fig. 5e and Supplementary Data 4). This particular variant is exclusively found within the lineage 4.3.1.2.1, which is known to be one of the ciprofloxacin-resistant H58 (4.3.1) lineages (Fig. 5f). The findings collectively indicate that there were convergent occurrences of hypo and hyper Vi capsule variations. Moreover, all instances of the hyper Vi capsule *tviD* A35S variant exhibit resistance to ciprofloxacin (Fig. 5f), which has traditionally been the mainstay treatment for patients with typhoid in clinical settings, which therefore suggests the potential for further dissemination of specific hyper Vi capsule variants.

## Discussion

Here, we performed pan-genome analyses of 5379 *S.* Typhi clinical isolates to determine whether the capsule variants of *S.* Typhi have emerged. Using the frequent clinical missense mutations in the *viaB* locus identified in this study, we conducted structural and functional studies to understand their consequences in *S.* Typhi pathogenicity and explain the molecular mechanisms involved. Resultantly, we generate a list of single missense mutations found in Vi capsule biosynthesis enzymes that are responsible for producing hypo and hyper Vi capsule variants of *S.* Typhi. This paper demonstrates the functional consequences of clinical missense mutations in the *S.* Typhi *viaB* locus. It serves to highlight the existence of hypo and hyper Vi capsule variants of *S.* Typhi, thereby increasing awareness in the community. These findings provide essential groundwork for enhancing preparedness in response to the potential future spread of hyper Vi capsule *S.* Typhi variants.

The existence of acapsular *S.* Typhi variants in human infection sites had been reported before this study. Acapsular *S.* Typhi variants lack the entire *viaB* locus or genes in the locus[50,51]. For instance, in Pakistan, 13 of the 102 clinical isolates[50] and 2 of the 35 clinical isolates[51] were found to be acapsular variants. Nothing was known regarding the emergence of other capsule variants. Here we discovered that three pathogenic variants of *S.* Typhi, the so-called 'hypo Vi capsule', 'intermediate Vi capsule', and 'hyper Vi capsule' variants, have different strengths in their pathogenicity. The 'hypo Vi capsule' *S.* Typhi variants have a markedly increased infectivity whose serum survival was almost comparable to WT *S.* Typhi. The 'hyper Vi capsule' *S.* Typhi variants are superb in evading the host innate immune system and colonizing the gallbladder and other organs, as a 100,000-fold increased bacterial burden was observed. The bacteria-associated hyper Vi capsule, with a likely contribution from the shed Vi, increases the infectivity of the hyper Vi capsule variants. It is also notable that the shed Vi can increase the infectivity of co-infected pathogens. The 'intermediate Vi capsule'

*S.* Typhi variants diverge into two subgroups: some variants phenocopy WT and the other phenocopy hyper Vi capsule variants.

The most striking finding of this study is that many of the frequent clinical missense mutations in *tviD* and *tviE* are hyper Vi capsule variants. Hyper Vi capsule *S.* Typhi strains were resistant to human neutrophil-mediated phagocytosis. In mice, hyper Vi capsule *S.* Typhi induced liver abscesses, clinical signs (overall weakness, eye discharge, and diarrhea), and deaths in 50% of infected mice. Intriguingly, all mice infected with hyper Vi capsule variants had *S.* Typhi colonized in their gallbladder on Day 3. Recent genome sequencing analyses of *S.* Typhi gallbladder isolates reported 3 of the hyper Vi capsule missense mutations that we found in this study: *tviD* R508H, *tviE* C137R, and *tviE* A462V[33,65], suggesting a possible correlation between hyper Vi capsule variants and heightened gallbladder colonization in humans.

The link between hypercapsule and hypervirulence was documented in other capsulated pathogens, including *Klebsiella pneumoniae*, *Acinetobacter baumannii*, and *Streptococcus pneumoniae*[6,67–71], which seems to point to parallel evolution in capsulated pathogens, even though their capsule specifics, including mucoidy, are different. The molecular bases for their hypercapsule phenotypes appear to be more complex than that of *S.* Typhi, as deletions, insertions, and several non-synonymous SNPs were involved[6,72–74]. The most similar evolution to that of *S.* Typhi is *K. pneumoniae*, as SNPs in the capsule biosynthesis gene *wzc* lead to hypercapsule production, which confers phagocytosis resistance, enhanced dissemination, and increased mortality in animal models[6,67]. In human infection sites, diarrhea and overall weakness are known general typhoid fever clinical signs. Although whether hyper Vi capsule *S.* Typhi was involved is unknown, typhoid fever cases with liver abscess, in some cases, with gallbladder colonization, have been reported[75–80].

We have made several technical advancements through the development and refinement of molecular tools and methodologies. We have created an in vivo molecular tool that enables us to examine the impact of clinical missense mutations on the mode of action for TviE. TviE studies, as well as our research utilized TviD C127R and two experimental mutations, illustrate the distinct correlation between amino acid substitutions found in clinical missense mutations and the hypo Vi capsule or hyper Vi capsule phenotypes observed in TviE and TviD variants. We note that the mechanistic studies on TviD are comparatively less comprehensive in comparison to those conducted on TviE. In contrast to TviE mutations, the mutations in TviD were observed to occur throughout its 3D structure, which presents challenges in identifying a discernible evolution theme. However, the complexity of this situation is consistent with the prediction that mutations affect various stages of Vi production. We predict that these stages include: (a) mutations that directly impact the catalytic activity, either inhibiting or promoting the reaction (e.g., C127R); (b) mutations that affect catalysis during the pre- or post-stage, involving the reception of the glycan substrate or the transfer of the resulting glycan to the next enzyme; and (c) mutations that influence the formation of multimers or multienzyme complexes, typically located on the surface and likely through the TPR domains. Further investigations are warranted to fully understand these effects by specific clinical mutations. As part of this study, we have developed a WGS analysis pipeline for *S.* Typhi, which is expected to be useful in the research community. Furthermore, it is expected that the murine model used, combined with the hyper Vi capsule *S.* Typhi, will be a valuable *S.* Typhi infection model for the initial phase of infection. This model will facilitate the detection of variations in virulence during the early stages of infection. However, it is crucial to recognize the necessity for additional research utilizing more appropriate animal models in order to investigate the effects of capsule variants during the later stages of infection.

Limitation of this study: We note that the SDS-PAGE migration and immunoblot-based assessments of Vi intensity and length have certain limitations. Changes in Vi chemistry, including *O*-acetylation, can

influence these assessments. Therefore, we should interpret the Vi immunoblot results as reflecting approximate, relative length and intensity differences, rather than precise measurements.

In summary, by combining epidemiology, genomics, and molecular investigation, here we provide much-needed valuable insights into the various types and consequences of the Vi capsule variants, as well as a future outlook on the occurrence and dissemination of the capsule variants of *S*. Typhi.

## Methods

### Ethics statement
Mouse studies were conducted according to a protocol approved by Cornell IACUC (protocol number 2014-0084). The process of collecting human blood samples was carried out in accordance with the protocol that has been approved by the Institutional Review Board for Human Participants at Cornell University (protocol number IRB0008664). A nurse practitioner at the Cornell Human Metabolic Research Unit conducted a peripheral blood draw to obtain primary neutrophils and sera. The data were analyzed in an anonymous manner. All adult participants (18–65 years old) provided informed consent. The written consent form was provided to all participants.

### Bacterial strains
*S*. Typhi ISP2825[52] was used as a parent strain for generating the following strains: *ΔtviBC*, *ΔtviD*, *ΔtviE-vexE* (a deletion of *tviE*, *vexA*, *vexB*, *vexC*, *vexD*, and *vexE*), *tviD* A35S, *tviD* C127R, *tviD* E236K, *tviD* H253A, *tviD* T293A, *tviD* T424I, *tviD* R508H, *tviD* R523H, *tviD* G616S, *tviD* S676N, *tviD* G752C, *tviD* D763N, *tviD* E810K, *tviE* C137R, *tviE* Q219R, *tviE* P263S, *tviE* K266N, *tviE* K266R, *tviE* K266E, *tviE* S290G, *tviE* S290R, *tviE* A421T, *tviE* A462V, *tviE* V495A, *tviA* V62I, *tviD* C127S, *tviD* C127A, WT (Cm^R), *tviE* P263S (Kan^R), and *tviE* K266N (Kan^R) (Supplementary Table 4). We note that we used gene names (e.g., *tviD*) with the protein code (e.g., C to R) and protein residue location (e.g., 127) to reflect missense mutations (and distinguish them from synonymous single-point mutations [SNPs]). Genetic manipulations were performed as described previously[59,81]. In brief, pSB890 (Tet^R with *sacB* for sucrose selection), a suicide vector, was used as a backbone to introduce point mutations and epitope tags. The vector pSB890 was digested with restriction enzymes (BamHI-HF [NEB, cat# R3136S] and NotI-HF [NEB, cat# R3189S]). Inserts for each mutant were amplified by conducting PCR reactions using Herculase® II Fusion DNA Polymerase (Invitrogen, cat# 600679) and specific primers with the *S*. Typhi ISP2825 genome as a template. The digested vector and inserts were Gibson-assembled (T5 exonuclease, NEB cat# M0363S; Phusion polymerase, NEB cat# M0530S; Taq DNA ligase, NEB cat# M0208L) to generate the required plasmids. The resulting plasmids were transformed into *E. coli* ß2163 Δnic35 for conjugation and subsequent homologous recombination in *S*. Typhi. All strains were verified by performing Sanger sequencing available through the Cornell Institute Biotechnology Resource Center (BRC) Genomics Facility (RRID: SCR_021727) or Eton Bioscience (RRID: SCR_003533).

### Mammalian cell culture conditions
Human epithelial cell Henle-407 was cultured in DMEM high glucose (Invitrogen) supplemented with 10% FBS (Hyclone cat# SH30396.03, Lot# AD14962284). Sialic acid contents of the FBS used were validated, which was ~99% Neu5Ac and less than 1% Neu5Gc. Cells were kept at 37 °C in a cell culture incubator with 5% CO$_2$. Mycoplasma testing was conducted regularly as part of the cell maintenance practice.

### *S*. Typhi whole genome sequence analyses
A bioinformatics pipeline was established for WGS comparison analyses of 5379 *S*. Typhi clinical isolates. In brief, all *S*. Typhi WGSs that were available in the NCBI's GenBank database (5749 strains as of

11/03/21) were re-annotated using the Prokaryotic Genome Annotation Pipeline (PGAP, 2021-07-01.build5508; RRID: SCR_021329) to synchronize all the annotations under same criteria. RefSeq's gene numbers were used to identify the gene pool from each strain and a single digit code was allocated to each gene: −1, absence of gene; 0, predicted pseudogene; 1, presence of gene, using a bash script. Digit code was left blank if the gene was partially sequenced (Supplementary Data 1). To ensure that the dataset only includes *S*. Typhi WGSs, the following two additional filters were used: 1) SeqSero2[46] and 2) typhoid toxin subunits (*cdtB*, *pltA*, and *pltB*)[47], resulting in a total of 5379 *S*. Typhi strains. Point mutations on each Vi synthesis gene were examined by using Clustal Omega (RRID: SCR_001591) multiple sequence alignments and visualized with Prism 9.4.1 (GraphPad, RRID: SCR_002798). WGS accession numbers and the digit codes of target genes used in this study are described in Supplementary Data 1.

For epidemiological studies, *S*. Typhi WGSs from the NCBI's GenBank database (5749 strains as of 11/03/21) and the Pathogenwatch (13,081 strains as of 12/27/23) were combined, and duplicates were removed. The *S*. Typhi WGSs of 16,368 strains were analyzed by genotyphi (https://github.com/typhoidgenomics/genotyphi) to define their lineages and antimicrobial resistance of ciprofloxacin and azithromycin. *tviE* and *tviD* nucleotide sequences were extracted using SeqKit[82], and single nucleotide polymorphisms (SNPs) of *tviE* and *tviD* were identified with a custom bash script. The data were visualized with Prism 9.4.1 (GraphPad, RRID: SCR_002798).

### Immunoblots
*S*. Typhi ISP2825 was grown in Luria-Bertani (LB) broth at 37 °C overnight. Fifty microliters of the culture were inoculated in 2 ml of 86 mM NaCl LB, and cultured until OD$_{600}$ reaches 0.9–1.0. A volume equivalent to OD$_{600}$ = 1.0 was pelleted at 5000 × g for 5 min and washed with PBS. When the shed Vi was indicated, 0.5 ml of the culture supernatant was saved after the aforementioned pelleting, which was further centrifuged at 20,000 × g for 10 min. Two hundred microliters of the supernatant were collected; 40 µl of 6× SDS sample buffer (360 mM Tris-HCl, 12% SDS, 60% Glycerol, 5% ß-mercaptoethanol, and 0.2% Bromophenol blue) containing 20 µg/ml proteinase K was added. Note that we used proteinase K for glycan immunoblots. When the Vi expression of *S*. Typhi was compared to the shed Vi in the supernatant, the bacterial pellet was resuspended in 1 ml of 1× SDS sample buffer containing 20 µg/ml proteinase K, followed by vortexing at 15-min intervals for 1 h at 37 °C. In most experiments that did not need to quantify the amounts of the shed Vi, the bacterial pellet was resuspended in 0.4 ml of 1× SDS sample buffer containing 20 µg/ml proteinase K. For protein immunoblots (Flag), the pellets were lysed in 0.4 ml of 1× SDS sample buffer without proteinase K, followed by boiling at 98 °C for 5 min. The lysates were centrifuged at 20,000 × g for 10 min; 5 µl (1 × 10^7 cells when 0.4 ml of 1× SDS buffer was used) of the lysates was run on 8–16% Mini-PROTEAN® TGX™ Precast Protein Gels (Bio-Rad, cat# 4561106) for Vi and O9 detection or 8% Tris-Glycine SDS-PAGE gel for Flag detection. The SDS-PAGE gels were wet-transferred at 20 V overnight at 4 °C using Bjerrum Schafer-Nielsen transfer buffer (48 mM Tris, 39 mM glycine, and 20% methanol) to nylon membranes (Biodyne B Nylon Membrane, Pall) for Vi and LPS or nitrocellulose membrane (Amersham, cat# 10600002) for protein samples. *Salmonella* Vi antiserum (pan-Vi antibody) at 1/300 dilution (BD Difco, cat# 228271, lot# 1357166, RRID: AB_2934033), anti-Vi monoclonal antibody (*O*Ac-Vi antibody) at 1/200 dilution (SSI-Diagnostica, cat# REF15514, lot# 188O-3, RRID: AB_2934031), or rabbit anti-O9 antiserum at 1/200 dilution (SSIDiagnostica, cat# REF40276, lot# 636O-H5, RRID: AB_2934032) was used to detect Vi or O9 antigens. SuperSignal™ West Femto Maximum Sensitivity Substrate (Thermo-Fisher, cat# 34095) was used to develop the images using iBright™ CL1500 Imaging System (ThermoFisher). Vi and O9 intensities were

calculated using iBright™ analysis software (ThermoFisher, RRID: SCR_017632), and quantified using Prism 9.4.1 (GraphPad, RRID: SCR_002798).

## Semi-RT-qPCR
One µg of total RNAs was reverse-transcribed with Maxima H minus reverse transcriptase (Thermofisher, cat# EP0752) using random hexamer, and the cDNA library was 10-fold diluted with MilliQ water. cDNA library was amplified with gene-specific primers (Supplementary Table 5) for 25 cycles except for 16S rRNA which was amplified for 16 cycles using Green Taq DNA Polymerase (Genscript, cat# E00043).

## TviE and TviD structure predictions
TviE or TviD structure was predicted using RoseTTAFold[56] and AlphaFold v2[55] or AlphaFold v2, respectively. Mapping the clinical missense point mutations, electrostatic potential calculation, and superimposition of the predicted TviE and TviD structures from AlphaFold v2 were performed using ChimeraX (University of California at San Francisco, RRID: SCR_01587).

## TviE-inducible system
**TviE-inducible system construction.** *S.* Typhi ISP2825 lacking *tviE-vexA-vexB-vexC-vexD-vexE* (Δ*tviE-vexE*) was constructed and transformed with pBAD-*tviE*-Flag carrying *tviE* sequence of WT, P263S (hyper Vi), K266N (hypo Vi), or S290R (hyper Vi) (Supplementary Table 4).

**TviE mutant characterization.** *S.* Typhi ISP2825 Δ*tviE-vexE* (pBAD-*tviE*-Flag) was cultured overnight in LB containing 100 µg/ml ampicillin and 0.5% glucose. Two hundred fifty microliters of the overnight culture were transferred into 10 ml of LB containing 86 mM NaCl, 100 µg/ml ampicillin, and 0.5% glucose and cultured until $OD_{600nm}$ reaches 0.9–1.0. A control sample (at 0 min, without L-arabinose induction) was collected at this step (a volume equivalent to $OD_{600nm} = 1$). The culture was centrifuged at $5000 \times g$ at 4 °C for 10 min and resuspended in 9 ml of LB containing 86 mM NaCl, 100 µg/ml ampicillin, and 0.1% L-arabinose; the culture was aliquoted into four tubes (2 ml each) and cultured. A volume equivalent to $OD_{600nm} = 1$ was collected and pelleted at each indicated time point (15, 30, 45, and 60 min). Immunoblots were performed as described above.

## Bacterial growth studies
The overnight culture was 10-fold serial-diluted, and further diluted in PBS to prepare $8 \times 10^6$ cells/ml (corresponding to $OD_{600nm} = 0.01$). Ten microliters of the prepared dilution ($8 \times 10^4$ cells) were added to 150 µl of 86 mM NaCl LB or 300 mM NaCl LB on a 96-well plate; $OD_{600nm}$ was measured at 37 °C at 30 min intervals for 24 h using Tecan Infinite 200 Pro microplate reader (Tecan); the plate was shaken for 5 s before each measurement.

## Image cytometry for Vi and O9 expressions
*S.* Typhi ISP2825 and the indicated each single amino acid point mutant was grown in LB overnight; 50 µl of the overnight culture was inoculated into 2 ml of 86 mM NaCl LB and cultured until $OD_{600nm}$ reaches 0.9–1.0. A volume equivalent to $OD_{600nm} = 1.0$ was pelleted at $5000 \times g$ for 5 min and resuspended in PBS. The suspension was pelleted at $5000 \times g$ for 5 min, re-suspended in PBS containing 4% paraformaldehyde (PFA), and incubated for 1 h. The lack of viability of *S.* Typhi was confirmed by spreading and culturing the fixed cells on LB plates. In parallel, during the fixation step, coverslips were coated with poly-D-lysine (Gibco, cat# A3890401) for 1 h, washed with Milli-Q water, and dried. In one hour, the fixed cells were pelleted at $5000 \times g$ for 5 min and re-suspended in 100 µl of PBS. The samples were placed on a poly-D-lysine-coated coverslip for 30 min; the coverslip was washed with PBS, incubated in PBS containing 3% BSA for 30 min for blocking,

replaced the blocking buffer with PBS/3% BSA containing *O*Ac-Vi antibody (1:200) and O9 antibody (1:200), and incubated for 1 h. In one hour, the coverslip was washed with PBS, replaced the buffer with PBS/3% BSA containing Alexa-488 goat anti-rabbit (Invitrogen, cat# A11034, RRID: AB_2576217, 1:4000) and Alexa-594 goat anti-mouse antibody (Invitrogen, cat# A11032, RRID: AB_2534091, 1:4000), and incubated for 30 min. Using a BZ-X810 (Keyence) microscope, *S.* Typhi was imaged under Plan Apochromat 60X Oil objective (BZ-PA60); we set the Keyence imaging program to automatically acquire and stitch 49 images (7 by 7), resulting in one overall image covering the 1.256 mm × 0.942 mm area of each coverslip. Alexa-Flour 594 brightness (Vi) of each of the acquired overall images was quantified; we let the BZ-X810 image analyzer find the area that had *S.* Typhi O9 staining and quantified the brightness for Vi. We note that the +5 expand option in BZ-X810 Analyzer was used given the location of Vi and LPS O9 on the surface of *S.* Typhi. The total Alexa-Flour 594 brightness (Vi) was then divided by the number of *S.* Typhi counted from O9 staining to determine the average Vi brightness/cell from the stitched image. All image cytometry processes of these fixed cells were performed at room temperature.

## Alcian blue staining of SDS-PAGE
Alcian blue staining of the SDS-PAGE gel was performed as described elsewhere. Briefly, SDS-PAGE gel was fixed with fix/wash buffer (25% ethanol/10% acetate) for 1 h and stained with fix/wash buffer containing 0.125% alcian blue for 15 min at 50 °C. The gel was then washed with a fix/wash solution overnight at room temperature.

## Mucoidy tests
*S.* Typhi was grown in a blood agar plate (tryptic soy agar with 5% defibrinated sheep blood, Cat# R111-0100, Rockland) or LB agar plate for overnight at 37 °C incubator. A sterile inoculation loop was used to test mucoidy. Strings less than 2 mm were determined to be negative.

## Invasion and adhesion assays
Henle-407 cells ($5 \times 10^4$) were seeded on a 24-well plate a day before infection. For imaging intracellular bacteria, a coverslip was placed on 24 well plates before seeding the cells. The cells were infected with *S.* Typhi WT or mutant in Hanks' Balanced Salt Solution (Invitrogen) at 15 multiplicity of infection (m.o.i.) for 1 h, washed with PBS, and incubated with 100 µg/ml gentamicin for 45 min to kill extracellular bacteria.

For colony forming unit (CFU) determination at 2 h post-infection, the infected cells were washed with PBS, lysed in 1 ml of PBS containing 0.1% sodium deoxycholate for 15 min at room temperature, and 100 µl of $10^{-1}$-diluted lysate was spread on LB agar plates. The cells for the 24 h post-infection were maintained in complete DMEM containing 10 µg/ml gentamicin. Colonies were counted to calculate the total number of intracellular bacteria.

For immunofluorescence assays, the coverslips were fixed in PBS/4% PFA overnight at 4 °C, washed with Tris-buffered saline (TBS) to quench PFA, and permeabilized for 30 min with PBS containing 3% BSA, 0.2% Triton X-100, and 10 mM Tris. For imaging intracellular *S.* Typhi after 2 h post-infection, the coverslips were incubated with anti-*Salmonella* antibody (Difco, cat# 240993, 1:4000) and Alexa-Fluor 488 Phalloidin (Invitrogen, cat# A12379) for 1 h, washed with PBS-T (PBS/0.1% Tween 20), incubated with Alexa Fluor-594-labeled anti-rabbit antibody for 30 min, and counterstained the nuclei with 4′,6-diamidino-2-phenylindole (DAPI, Invitrogen, cat# D3571) for 10 min. Intracellular *S.* Typhi was imaged using a BZ-X810 microscope (Keyence); Plan Fluorite 20X LD PH (BZ-PF20LP) objective was used to acquire 100 images covering 5.290 mm × 3.968 mm area of each coverslip. We let the BZ-X810 image analyzer find host cells (Phalloidin+) and count intracellular *S.* Typhi; the total counted number of *S.* Typhi was divided by the number of DAPI-stained nuclei to determine *S.* Typhi/cell from

each image. The fully-focused images presented in Fig. 3i were acquired using Plan Apochromat 60X Oil (BZ-PA60) objective.

For inside-outside staining presented in Fig. 3k–l (1 h post infection), the fixed cells on coverslips were incubated with PBS/3% BSA for 30 min before incubating with anti-*Salmonella* antibody for 2 h to stain extracellular, attached bacteria. The coverslips were washed with PBS and incubated with Alexa Fluor-488-labeled anti-rabbit antibody for 1 h. The coverslips were then washed with PBS and permeabilized for 30 min with PBS containing 3% BSA, 0.2% Triton X-100, and 10 mM Tris. To stain both intracellular and extracellular, attached bacteria, the coverslips were incubated with anti-*Salmonella* antibody for 1 h, washed with PBS-T, incubated with Alexa Fluor-594-labeled anti-rabbit antibody for 1 h, and the nuclei were counterstained with DAPI. All the processes were performed at room temperature except PFA fixation.

### Image acquisition and quantification

Immunofluorescent and agglutination images were acquired by using a BZ-X810 microscope (Keyence). The filters used in this study are as follows: Alexa-Fluor 488 and Alexa-Fluor 488 Phalloidin, 470/40 nm excitation with 525/50 nm emission (OP-87763); Alexa-Fluor 594, 560/40 nm excitation with 630/75 nm emission (OP-87765); DAPI, 360/40 nm excitation with 460/50 nm emission (OP-87762). The immunofluorescent images of intracellular *S.* Typhi were quantified with the 'Hybrid Cell Count' function of the BZ-X810 image analyzer. For quantifying *S.* Typhi aggregates in agglutination assays, CellProfiler Image Analysis Software (BROAD institute, RRID: SCR_007358) was utilized. For Vi quantification, a built-in image acquisition function of the iBright CL1500 imager (ThermoFisher) acquired a set of images with varied exposure times. This image acquisition function is programmed to find an optimal exposure time for a maximum signal but below saturation, as the signal saturation is displayed as a red colored band(s) on the acquired images. We only used images that showed no signs of red-colored saturation for Vi quantification. The quantification was conducted using the iBright analysis software 5.1.0 (ThermoFisher, RRID: SCR_017632). The differences of Vi length were analyzed using ImageJ 1.53k (National Institutes of Health, RRID: SCR_003070). All statistics were performed using Prism 9.4.1 (GraphPad, RRID: SCR_002798). For the relative migration quantification, the data was acquired using ImageJ's line profiling with the Lorentzian (Cauchy) model to get a center value, which was used for determining the values of relative migration.

### Serum resistance assay

**Human sera.** Both males and females aged 18–65 years donated their blood samples; however, we did not record their biological sex information to conduct this study in a de-identified manner. De-identified human blood samples were clotted for 1 h at room temperature, and centrifuged at 2000 × g. The supernatant (healthy human serum) was harvested, aliquoted, and stored at −80 °C until use. When indicated, a serum aliquot was incubated at 56 °C for 1 h to prepare complement-inactivated human serum (iHS).

### Serum resistance assay

Overnight *S.* Typhi culture was prepared as described above; 50 μl of the culture was inoculated into 2 ml of 86 mM NaCl LB and incubated at 37 °C until $OD_{600nm}$ reaches 0.9–1.0. A volume equivalent to $OD_{600nm} = 1.0$ was centrifuged at 5000 × g for 5 min. The pellet was resuspended in 0.5 ml PBS, after which 10 μl of *S.* Typhi ($1.6 \times 10^7$ cells) was added into 150 μl PBS ($1.6 \times 10^7$ cells/160 μl). For serum resistance assays, 10 μl of *S.* Typhi in PBS ($1 \times 10^6$ cells) was added into 90 μl of PBS in the absence and presence of human serum or inactivated human serum. The samples were incubated for 2 h at 37 °C. CFU was determined by culturing 10-fold serial dilutions on LB agar plates; colonies were counted after 16 h. Plate images were acquired using an iBright™ CL1500 Imaging System (ThermoFisher).

### Vi purification

*S.* Typhi WT, Δ*tviBC*, or *tviE* P263S was grown in LB broth at 37 °C overnight. Two hundred fifty microliters of the culture were inoculated in 10 ml of 86 mM NaCl LB, and cultured until $OD_{600}$ reaches 1.5. The culture was centrifuged at 2000 × g for 20 min at 4 °C. The supernatant of each strain was 0.22 μm-filtered, and incubated with DNase I and RNase A for 2 h at 37 °C, followed by incubating with Proteinase K for 2 h at 37 °C. The supernatant (10 ml) was 0.22 μm-filtered before being concentrated to 0.5 ml using a 30 kDa cut-off filter. The resulting 0.5 ml was mixed with 10 ml PBS before being passed through the 30 kDa cut-off filter to bring the volume to 0.5 ml. This washing-concentration procedure was repeated 6 times to ensure that the enzymes and other contaminants were removed. The resulting samples (100 μl) were incubated with anti-O9-conjugated agarose beads for 1 h at RT to minimize LPS contamination. The Vi was further-purified with size exclusion chromatography (AKTA) using Superdex 200 Increase 10/300 GL column (Cytiva, cat# 28-9909-44), and quantified by conducting immunoblots.

### Animal experiments

All infection experiments were conducted in a clean SPF ABSL-2 environment within the TMCF animal facility. This room in the TMCF follows a 10-14 light/dark cycle. The lights turn on at 5 a.m. and off at 7 p.m. The temperature range is 65–75 °F (18–24 °C), with humidity levels ranging from 30 to 70%. Food and water were provided ad libitum.

Cmah-null mice expressing the human-type glycan receptor (Jackson Laboratory, RRID: IMSR_JAX:017588)[47,57,59,60,63,83–85] were infected intraperitoneally with $8 \times 10^5$ *S.* Typhi WT, *tviE* K266N, or *tviE* P263S strain in 100 μl of PBS. Both males and females aged 6- to 12-weeks were used, and no sex-dependent phenotypic differences with regards to *S.* Typhi susceptibility were observed. Infected mice were sacrificed, and organs (liver, spleen, and gallbladder) were harvested 3 days after infection, except two mice in *tviE* P263S group that died on Day 1. The organs were mechanically homogenized with PBS/0.05% sodium deoxycholate; 5 ml, 3 ml, or 2 ml of lysis buffer was used for the liver, spleen, and gallbladder, respectively. CFUs were determined by plating 10-fold serial dilutions of homogenates on LB agar plates. The number of total CFU in each organ was calculated by multiplying dilution factors.

For co-infection experiments of WT with *tviE* P263S or *tviE* K266N, $4 \times 10^5$ WT *S.* Typhi (Chloramphenicol-resistant) and $4 \times 10^5$ *S.* Typhi *tviE* P263S (Kanamycin-resistant) or K266N (Kanamycin-resistant) were prepared in 100 μl of PBS ($8 \times 10^5$ in total) for infection. Organ CFUs were assessed by plating the diluted lysates on LB agar plates containing 2.5 μg/ml Chloramphenicol for WT or 50 μg/ml Kanamycin for *tviE* P263S and K266N. For WT *S.* Typhi infection experiments with shed Vi, the amount of shed Vi corresponding to the surface Vi of ~1.6 × $10^9$ *S.* Typhi was mixed with $8 \times 10^5$ *S.* Typhi WT in 100 μl of PBS for infection. *S.* Typhi Δ*tviBC* preparations having comparable LPS concentrations were used as a control.

### Measurement of ROS burst from human peripheral blood neutrophils

Human peripheral blood neutrophils from human blood samples were isolated as described elsewhere. Briefly, 10 ml of peripheral blood were collected in a EDTA-treated tube, and gently mixed with 5 ml of 3% gelatin (Difco, cat# 214340)/0.1% glucose/0.9% NaCl. The red blood cells were sedimented at RT for 45 min. The neutrophil-rich supernatant was transferred onto the top of 3 ml of Ficoll-Paque Premium (GE Healthcare, cat# 17-5442-02) in a 15 ml conical tube. The samples were centrifuged at 280 × g for 20 min at 16 °C. The pellet was resuspended in 6 ml of red blood cell lysis buffer (1 ml of PBS and 5 ml of distilled water) and incubated for 45 s, followed by adding 2 ml of 3% NaCl solution to stop the reaction. The neutrophil samples were

centrifuged at $280 \times g$ for 5 min at 4 °C, and resuspended in 1 ml of 2% FBS/phenol red-free RPMI-1640. *S.* Typhi WT and capsule variants were grown in 2 ml of 86 mM NaCl LB until $OD_{600}$ reaches 1.0. Each strain was opsonized with human sera in PBS for 30 min at RT. $5 \times 10^4$ neutrophils in 90 μl of phenol-free RPMI (ThermoFisher, cat# 11835030) supplemented 2% FBS with 1 mM luminol (Sigma, cat# 123072) were seeded into a black opaque 96-well plate, and 10 μl of opsonized *S.* Typhi ($5 \times 10^5$ in 10 μl) were added (10 m.o.i.). Luminescence was recorded at 2-min interval using Tecan Infinite 200 Pro microplate reader (Tecan).

## Quantification and statistical analysis
Data were tested for statistical significance with the GraphPad Prism software. The number of replicates for each experiment and the statistical test performed are indicated in the figure legends. Image analysis and quantification were performed using ImageJ. The number of biological replicates and the statistical method are described in each figure legend. At least 3 independent experiments were performed throughout the study.

## Reporting summary
Further information on research design is available in the Nature Portfolio Reporting Summary linked to this article.

## Data availability
The published article includes all datasets generated during this study. WGS accession numbers and the digit codes of target genes used in this study are described in Supplementary Data 1. Source data are provided with this paper.

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

## Acknowledgements

We thank the members of the Song laboratory for engaging in thoughtful discussions throughout the study. Parts of this work were supported by NIH AI139625 and AI141514 to J.S. The funders had no role in the study design, data collection, analysis, decision to publish, or preparation of the manuscript.

## Author contributions

G.Y.L. conceptualized this research, executed all experiments, interpreted the results, and wrote the manuscript. J.S. conceptualized this research, acquired the funding, interpreted the results, executed animal experiments, and wrote the manuscript. Competing interests: A provisional patent application has been filed for parts of this work (J.S. and G.Y.L.).

## Competing interests

The authors declare no competing interests.
