## [Peer Review File · Nature Communications]

Single missense mutations in Vi capsule synthesis genes confer hypervirulence to *Salmonella* TyphiEditorial Note: This manuscript has been previously reviewed at another journal. This document only contains reviewer comments and rebuttal letters for versions considered at *Nature Communications*.

REVIEWER COMMENTS

Reviewer #1 (Remarks to the Author):

Lee et al revise their recently submitted manuscript on the discovery of hypervirulent *Salmonella* Typhi. Authors now include alcian blue binding as an additional method to measure capsule production in hypocapsule mutants, investigate capsule shedding, perform confirmational infection experiments and include more epidemiology data.

Major concerns

Previous concerns from this reviewer have not been adequately addressed:

- Capsule phenotypes are insufficiently characterized. This study is about the impact of adaptive mutations on capsule production in *S. Typhi*. Thus, a thorough characterization of the capsule phenotypes shown in Fig 2b should be the basis of this manuscript. Electron microscopy could confirm figure 2b and remove any doubt on structure possibilities. India ink staining of bacteria could at least show that the capsule looks bigger/longer. Alcian blue staining via SDS Page could confirm the antibody based binding and perhaps better demonstrate higher molecular weight capsule. If the bacteria look different as presented in 2b, the authors should demonstrate it. The authors make a case for increased length with the *tviE* mutants, which do show a higher smear in the immunoblot, but this is not the case for most of the *tviD* hypercapsule mutants, which show increased intensity of the smear but not at higher molecular weights. Only the *tviE* mutants displaying a smear at increased molecular weights are shown in the two dimensional plots. In light of the novel data on shedding, one might think that increased capsule production in the *tviD* mutants just leads to increased shedding not necessarily to hypercapsule as shown in 2b. The alcian blue staining that the authors now provide in the revised version of the manuscript is only shown as a plotted figure without any indication of molecular weight and more importantly, it only addresses the hypocapsule mutants, not the hypercapsule mutants which are the biggest concern in terms of structure of capsule. Given the wide smear seen with anti Vi antibodies an alternative method of quantifying and characterizing capsule with more hypercapsule types (increased production, increased production+elongation) is needed to confirm or expand figure 2b, and the phenotype has to be visualized (EM or india ink). Authors also have to address mucoidity, as it is a hallmark of hypercapsule production and could be useful for the detection of these strains.

- Distinguishing supercapsule vs hypercapsule. Authors still provide no rationale for distinguishing supercapsule and hypercapsule mutants. "Based on obtained results" is not a rationale (l1123). This seems to be a consequence of the insufficiently characterized capsule phenotypes. The intermediate phenotype looks very similar to the hypercapsule phenotype (*tviD* mutants in Fig. 1i). Different types of hypercapsule (overall production vs elongation) may have different virulence phenotypes and warrant a distinction instead of the classification of "supercapsule" of mutants with intermediate phenotypes.

- Capsule excretion in hypercapsule mutants. Excretion data is presented as crucial for the virulence. However, in this setting, adding purified cps to the WT increases infectivity to a level that is otherwise not seen (10^6 vs 10^4). While this provides an indication that shedding of capsule from mutants can increase infectivity in this artificial setting it does not generally show that it is crucial for the virulence observed in mutant strains. The experiment shows that shedding a hypercapsule may contribute to the observed virulence of the mutants but this would have to be proven with a mutant that is not able to shed, which seems impossible. thus, l1308-310, 384-386 are an overstatement.

Minor concerns

- Processivity of TviE. Authors decide to expand this mostly hypothetical part. The assay and mutants that they describe do not provide more insight onto the role of the mutations compared to Fig. 1 (general role of the mutations on cps production). Most of this part should be in the discussion. The model should just briefly be introduced in the results section because of the limitations of the data. It is otherwise a distraction.

- Hypocapsule mutant phenotype. While the authors emphasize that hypocapsule mutants have no fitness cost, they do show increased phagocytic activity of hypocapsule mutants (4m). Increased phagocytosis is a classic phenotype of hypocapsule mutants.

- II.54-55 "Vi promotes macrophage phagocytosis" authors probably mean prevent?

-1199 authors should be more careful with the fitness cost statement, they only looked at growth in axenic medium.

Reviewer #2 (Remarks to the Author):

The authors have adequately addressed my comments regarding a previous version of the manuscript.

Reviewer #4 (Remarks to the Author):

I did not review the original submission and was asked to comment on the authors' responses to reviewer 3. I believe the authors have appropriately addressed the biological questions but I have concerns about some of the more biochemically-focused aspects. I evaluated the manuscript before looking at that reviewer's comments and it is clear we have similar concerns, several of which are not resolved in the revised version.

Overall, I find the observations in this manuscript very interesting. The phenotypes described here are fascinating, the connection to pathogenesis strong, and the bioinformatics analysis is comprehensive. However, some conclusions concerning enzyme functions are not properly supported by data or discussed precedent. These sections suffer from overinterpretation and/or lack of clarity and weaken the presentation.

1. The biosynthetic scheme and text description suggests an order of events that is not proven. For example, it is unknown whether Vi is extended at the reducing or non-reducing end of the chain (see <https://doi.org/10.1016/j.jbc.2022.102520>), and the addition of the terminal glycolipid could be the first or last step.

2. The distinction between hyper- and super-capsule variants is arbitrary and overly complicated. Given the range of apparent size distributions, I would recommend simplifying this to use only hypo- and hyper- group designations.

3. Figures 1j, 2i and 2j show plots with "Relative MW". I am not sure what the units are – are these kDa? The data is derived from westerns, and it is not possible to assign MW for a polysaccharide based on PAGE migration. If these values are kDa, then I assume they are based of protein standards whose migration in PAGE, which can only serve as a standard for migration for polysaccharides. I think it is fine to refer to "relative migration" as a correlate of apparent size but no more than that.

4. Signals from western blots are used quantitatively ("Vi intensity"). This assumes that all blots

are corrected for saturation and usually requires multiple runs and exposure times. Was this done? It also requires no changes in epitopes which may not be the case here if O-Ac content fluctuates. In general, I think the altered size distribution of Vi in the *tvIE* mutants is compelling but changes in the OAc-Vi blots less so. Data for acetylation would be more convincing with determined Ac:GalNAcA values but I recognize this can be challenging. Without this, the authors need to be much clearer about limits of interpretation. It would also be useful to see that the changes in cell bound material in all mutants are reflected in the released Vi by including corresponding western panels in this figure for direct comparison.

5. The interpretation of the *tvIE* mutations is confusing, often overstated, and lacks references to relevant literature:

I am not aware of data that definitively shows *TviE* is processive as claimed (line 129).

TviE is a GT4 enzyme and has been modelled before. It would be helpful to comment on the fit between this particular model and solved GT4 structures.

What is meant by "catalytic dyad"? The term is vague – what is the predicted role of these residues?

The data is interpreted as the "horizontal groove" being an acceptor binding site. This is one possible interpretation but what is known of acceptor binding in GT-B-fold enzymes to support this hypothesis? The proposal is highly speculative at this stage and the case is not built well.

The authors invoke an "ultrafast elongation processive mode" (line 143). There is no data on the "rate" of synthesis, nor does the phenotype require any change in rate per se.

What data supports the claim that a single residue change "is likely to be sufficient in altering the interaction dynamics between the enzyme and Vi elongation".

Figure 2d is confusing – why do the residue changes yield the shorter glycans that are shown? I am not sure what the authors intend to show here and without this being explained, the cartoon offers no added value. Also, why is there free GalNAcA monomer in the active site? It is supplied as UDP-GalNAcA.

I am not sure what is meant by "the first example of its kind" (line 170). GT4 enzymes are widespread.

6. With *TviD* the authors refer to gaining "a mechanistic understanding" (line 175) but no mechanistic data or critical analysis is presented. As the authors state, the distribution of mutations offers no insight, and the depth of understanding is not really enhanced by listing the possibilities.

I do not understand the statement about "adaptive evolution" and "high degree of specificity" (lines 190-192). Adaptation and specificity to what?

The claim that the C127 mutation must affect O-acetyl transfer is obvious. Are the authors proposing an affect on catalysis or protein folding or both. There is no data showing that this mutant actually results in stable protein in the cell. This seems vital for any interpretation.

Line 418 refers to "mechanistic studies conducted by (should be "on"?) *TviD*. Are the authors referring to past studies here?

Line 426 – what "loop region".

If the mutations affect *TviD* and E interaction, these should be in the TPR domains as described in ref 19.

7. In figure 4, the authors use "purified" Vi. I would use another term as the material isn't pure. How was the amount of added Vi determined and standardized given the different apparent size distributions.

Other -

Line 26 – what does "targeting" cover. I assume the authors refer to vaccines. The subsequent text suggests there are extensive examples of anti-capsule strategies but I am not sure this is true. There are a few successful examples.

Line 30 – UDP-N-acetyl-d-galactosaminuronic acid; the d should be a small capital D. Check for other examples. Also, this is an activated donor substrate – not a monomer as described.

Line 100 - is there data supporting the claim that *TviE* directly "controls" Vi length or is this an

assumption?

Line 120 – Measured transcripts are not a convincing marker for expression. The level of stable protein is the critical issue. I would refer here to the later data with “see below” to indicate overall protein levels seem unchanged.

Line 316-318 – there seems to be something missing from this sentence.

Line 410 – spelling - Klebsiella pneumoniaE

Point-by-point response to reviewers

We thank all reviewers for their comments and suggestions on the manuscript. Our point-by-point responses are as follows. Furthermore, new changes made in the revised version of the manuscript are highlighted in the manuscript (in red).

Reviewer #1

Lee et al revise their recently submitted manuscript on the discovery of hypervirulent *Salmonella* Typhi. Authors now include alcian blue binding as an additional method to measure capsule production in hypocapsule mutants, investigate capsule shedding, perform confirmational infection experiments and include more epidemiology data.

Major concerns

Previous concerns from this reviewer have not been adequately addressed:

- Capsule phenotypes are insufficiently characterized. This study is about the impact of adaptive mutations on capsule production in *S. Typhi*. Thus, a thorough characterization of the capsule phenotypes shown in Fig 2b should be the basis of this manuscript. Electron microscopy could confirm figure 2b and remove any doubt on structure possibilities. India ink staining of bacteria could at least show that the capsule looks bigger/longer. Alcian blue staining via SDS Page could confirm the antibody based binding and perhaps better demonstrate higher molecular weight capsule. If the bacteria look different as presented in 2b, the authors should demonstrate it.

The authors make a case for increased length with the *tviE* mutants, which do show a higher smear in the immunoblot, but this is not the case for most of the *tviD* hypercapsule mutants, which show increased intensity of the smear but not at higher molecular weights. Only the *tviE* mutants displaying a smear at increased molecular weights are shown in the two dimensional plots. In light of the novel data on shedding, one might think that increased capsule production in the *tviD* mutants just leads to increased shedding not necessarily to hypercapsule as shown in 2b. The alcian blue staining that the authors now provide in the revised version of the manuscript is only shown as a plotted figure without any indication of molecular weight and more importantly, it only addresses the hypocapsule mutants, not the hypercapsule mutants which are the biggest concern in terms of structure of capsule. Given the wide smear seen with anti Vi antibodies an alternative method of quantifying and characterizing capsule with more hypercapsule types (increased production, increased production+elongation) is needed to confirm or expand figure 2b, and the phenotype has to be visualized (EM or india ink). Authors also have to address mucoidity, as it is a hallmark of hypercapsule production and could be useful for the detection of these strains.

We value the feedback provided by this reviewer. It is important to note that Vi specific antibody-based methods have been widely accepted as the standard for characterizing the *Salmonella* Vi capsule in the field of *Salmonella* research (PMID: 34896394, 27226298, 29354293, and 36152747). As requested, we conducted alcian blue staining of SDS-PAGE for WT, acapsular, one hypo Vi capsular, and four hyper Vi capsular variants. Briefly, one-half of the cultures of the indicated *S. Typhi* strains were analyzed by alcian blue SDS-PAGE, while the remaining half of the samples were analyzed using immunoblots simultaneously. The molecular weight differences were similar in both results. We observed numerous non-specific background signals present in alcian blue-stained SDS-PAGE, in contrast to the specific detection of Vi observed in immunoblot results (S. Fig. 6, lines 128-133). Another question was about the two types of *tviD* mutants. In the revised manuscript, we conducted additional analysis to quantitatively assess the intensity, migration/length, and acetylation differences of Vi variants (Fig. 1h-k and S. Fig. 5). We made additional clarifications on hypo, super/intermediate, and hyper Vi variation (lines 117–124). Furthermore, as per request, we assessed the mucoidity of different hyper Vi capsule variants. We found that, unlike other bacteria, the hyper Vi capsule variation in *Salmonella* strains is not associated with mucoidity (S. Table 5, lines 137-141). To make sure that this work focuses on *Salmonella* Vi capsule variants, we have made a change in our manuscript. Instead of using 'hypo capsule' and 'hyper capsule', we now refer to them as 'hypo Vi capsule' and 'hyper Vi capsule'.

- Distinguishing supercapsule vs hypercapsule. Authors still provide no rationale for distinguishing supercapsule and hypercapsule mutants. "Based on obtained results" is not a rationale (l123). This seems to be a consequence of the insufficiently characterized capsule phenotypes. The intermediate phenotype looks very similar to the hypercapsule phenotype (*tviD* mutants in Fig. 1i). Different types of hypercapsule (overall

production vs elongation) may have different virulence phenotypes and warrant a distinction instead of the classification of "supercapsule" of mutants with intermediate phenotypes.

In the revised manuscript, we conducted additional analysis to quantitatively assess the intensity, migration/length, and acetylation differences of Vi variants (Fig. 1h-k and S. Fig. 5). We made additional clarifications on hypo, super/intermediate, and hyper Vi variation (lines 117–124).

- Capsule excretion in hypercapsule mutants. Excretion data is presented as crucial for the virulence. However, in this setting, adding purified cps to the WT increases infectivity to a level that is otherwise not seen (10e6 vs 10e4). While this provides an indication that shedding of capsule from mutants can increase infectivity in this artificial setting it does not generally show that it is crucial for the virulence observed in mutant strains. The experiment shows that shedding a hypercapsule may contribute to the observed virulence of the mutants but this would have to be proven with a mutant that is not able to shed, which seems impossible. thus, ll308-310, 384-386 are an overstatement.

We made an adjustment in the revised manuscript to say, "The bacteria-associated hyper Vi capsule, with a likely contribution from the shed Vi, increases the infectivity of the hyper Vi capsule variants." (lines 312-314 and line 389-390).

Minor concerns

- Processivity of TviE. Authors decide to expand this mostly hypothetical part. The assay and mutants that they describe do not provide more insight onto the role of the mutations compared to Fig. 1 (general role of the mutations on cps production). Most of this part should be in the discussion. The model should just briefly be introduced in the results section because of the limitations of the data. It is otherwise a distraction.

We made an adjustment to the revised manuscript regarding Fig. 2 result descriptions (lines 154-158).

- Hypocapsule mutant phenotype. While the authors emphasize that hypocapsule mutants have no fitness cost, they do show increased phagocytic activity of hypocapsule mutants (4m). Increased phagocytosis is a classic phenotype of hypocapsule mutants.

Like indicated below, we made an adjustment to the revised manuscript with regards to "fitness cost".

- ll.54-55 "Vi promotes macrophage phagocytosis" authors probably mean prevent?

In contrast to the role of Vi in the evasion of neutrophil phagocytosis, it has been shown that Vi promotes macrophage phagocytosis through the interaction between Vi and human C-type lectin DC-SIGN (PMID: 36286551). This seems counterintuitive, but it is beneficial to *Salmonella* (as *Salmonella* is equipped with its virulence factors that help thrive in macrophages). Given the focus of this work and its relevance to the referred description, we removed this particular aspect of *Salmonella* pathogenesis from the revised manuscript.

-ll199 authors should be more careful with the fitness cost statement, they only looked at growth in axenic medium.

We made an adjustment to the revised manuscript. The referred description now reads, "This suggests that there is no discernible growth defect associated with the production of variant forms of the Vi capsule in axenic cultures." (lines 209-210).

Reviewer #2:

The authors have adequately addressed my comments regarding a previous version of the manuscript.

Reviewer #4:

I did not review the original submission and was asked to comment on the authors' responses to reviewer 3. I believe the authors have appropriately addressed the biological questions but I have concerns about some of the more biochemically-focused aspects. I evaluated the manuscript before looking at that reviewer's comments and it is clear we have similar concerns, several of which are not resolved in the revised version.

Overall, I find the observations in this manuscript very interesting. The phenotypes described here are fascinating, the connection to pathogenesis strong, and the bioinformatics analysis is comprehensive. However, some conclusions concerning enzyme functions are not properly supported by data or discussed

precedent. These sections suffer from overinterpretation and/or lack of clarity and weaken the presentation.

1. The biosynthetic scheme and text description suggests an order of events that is not proven. For example, it is unknown whether Vi is extended at the reducing or non-reducing end of the chain (see <https://doi.org/10.1016/j.jbc.2022.102520>), and the addition of the terminal glycolipid could be the first or last step. “We note that the exact order of the Vi synthesis process remains to be characterized” has been added to Fig. 1 caption and lines 33 and 86.

2. The distinction between hyper- and super-capsule variants is arbitrary and overly complicated. Given the range of apparent size distributions, I would recommend simplifying this to use only hypo- and hyper- group designations. In the revised manuscript, we conducted additional analysis to quantitatively assess the intensity, migration/length, and acetylation differences of Vi variants (Fig. 1h-k and S. Fig. 5). We made additional clarifications on hypo, super/intermediate, and hyper Vi variation (lines 117–124).

3. Figures 1j, 2i and 2j show plots with “Relative MW”. I am not sure what the units are – are these kDa? The data is derived from westerns, and it is not possible to assign MW for a polysaccharide based on PAGE migration. If these values are kDa, then I assume they are based of protein standards whose migration in PAGE, which can only an serve as a standard for migration for polysaccharides. I think it is fine to refer to “relative migration” as a correlate of apparent size but no more than that. We have replaced “relative MW” with “relative migration” for the X-axis labels of Figures 1j-k and 2i-j. A relevant description is also added to the figure captions.

4. Signals from western blots are used quantitatively (“Vi intensity”). This assumes that all blots are corrected for saturation and usually requires multiple runs and exposure times. Was this done? It also requires no changes in epitopes which may not be the case here if O-Ac content fluctuates. In general, I think the altered size distribution of Vi in the *tviE* mutants is compelling but changes in the OAc-Vi blots less so. Data for acetylation would be more convincing with determined Ac:GalNAcA values but I recognize this can be challenging. Without this, the authors need to be much clearer about limits of interpretation. It would also be useful to see that the changes in cell bound material in all mutants are reflected in the released Vi by including corresponding western panels in this figure for direct comparison. Yes, that was done to conduct comparative immunoblot analyses. We included all three images that were used for comparative quantification analyses (S. Fig. 4). We value other comments that will be useful for future studies.

5. The interpretation of the *tviE* mutations is confusing, often overstated, and lacks references to relevant literature:

I am not aware of data that definitively shows *TviE* is processive as claimed (line 129). “processive” is removed (line 143).

TviE is a GT4 enzyme and has been modelled before. It would be helpful to comment on the fit between this particular model and solved GT4 structures. Additional biochemical characterizations are for future studies. We appreciate the reviewer’s comments for future studies.

What is meant by “catalytic dyad”? The term is vague – what is the predicted role of these residues? “dyad” has been removed and replaced with “residues”.

The data is interpreted as the “horizontal groove” being an acceptor binding site. This is one possible interpretation but what is known of acceptor binding in GT-B-fold enzymes to support this hypothesis? The proposal is highly speculative at this stage and the case is not built well. Additional biochemical characterizations are for future studies. We appreciate the reviewer’s comments for future studies.

The authors invoke an “ultrafast elongation processive mode” (line 143). There is no data on the “rate” of synthesis, nor does the phenotype require any change in rate per se. We removed “ultrafast elongation processive mode” from the referred description (lines 154-158).

What data supports the claim that a single residue change “is likely to be sufficient in altering the interaction dynamics between the enzyme and Vi elongation”. We modified this description to “In this possible scenario, we predict that the presence of a single amino acid variation in the groove is likely sufficient to alter Vi elongation” (line 157). This was based on the Vi immunoblots in Fig. 1 and 2.

Figure 2d is confusing – why do the residue changes yield the shorter glycans that are shown? I am not sure what the authors intend to show here and without this being explained, the cartoon offers no added value. Also, why is there free GalNAcA monomer in the active site? It is supplied as UDP-GalNAcA. I am not sure what is meant by “the first example of its kind” (line 170). GT4 enzymes are widespread. We made adjustments to Fig. 2d and removed “the first example of its kind”.

6. With TviD the authors refer to gaining “a mechanistic understanding” (line 175) but no mechanistic data or critical analysis is presented. As the authors state, the distribution of mutations offers no insight, and the depth of understanding is not really enhanced by listing the possibilities. “mechanistic” is removed and replaced with “insights into” (line 184).

I do not understand the statement about “adaptive evolution” and “high degree of specificity” (lines 190-192). Adaptation and specificity to what? “adaptive” and “high degree of specificity” were removed (lines 200-201). The claim that the C127 mutation must affect O-acetyl transfer is obvious. Are the authors proposing an affect on catalysis or protein folding or both. There is no data showing that this mutant actually results in stable protein in the cell. This seems vital for any interpretation. We would like to point out several observations. First, *tviD* C127R is a clinical missense mutation that circulates among human patients. Bacteria making a protein with a folding and stability issue are unlikely to survive for a long time. Second, if TviD C127R has a protein folding or stability issue, this *Salmonella* Typhi mutant should phenocopy *S. Typhi* Δ *tviD* (S. Fig. 1). However, these two mutants exhibit opposite phenotypes: hypo Vi capsule (Fig. 2l) and hyper Vi capsule (S. Fig. 1b), respectively. Third, the C127 and R127 residues seem to be well-aligned in TviD structures (rebuttal letter Fig. 1). Lastly, given the phenotypes of *S. Typhi* Δ *tviD* and *tviD* H253A (catalytically inactive) mutants (S. Fig. 1), we predict that the *tviD* C127R mutant affects the catalytic efficiency of TviD. However, we note that additional biochemical characterizations are for future studies (line 204).

Line 418 refers to “mechanistic studies conducted by (should be “on”?) TviD. Are the authors referring to past studies here? ‘conducted by’ has been replaced with ‘on’.

Line 426 – what “loop region”. Removed.

If the mutations affect TviD and E interaction, these should be in the TPR domains as described in ref 19. We add “likely through the TPR domains” to the referred description (line 433).

7. In figure 4, the authors use “purified” Vi. I would use another term as the material isn’t pure. How was the amount of added Vi determined and standardized given the different apparent size distributions. In the revised manuscript, we replaced “purified” with “preparations” (Lines 301-303). The amount of added Vi per mouse was corresponding to the Vi obtained from $\sim 1.6 \times 10^9$ *S. Typhi* strains indicated, and it was described in the Method section. We chose this bacteria-number based determination method as Vi diversity occurs at individual bacterial levels (e.g., WT vs. hyper Vi capsule variant). We note that all samples were handled simultaneously; if there was any loss or impurity during the purification procedures, this should occur across all samples.

Other -

Line 26 – what does “targeting” cover. I assume the authors refer to vaccines. The subsequent text suggests there are extensive examples of anti-capsule strategies but I am not sure this is true. There are a few successful examples.

‘(e.g., vaccine)’ is added after “targeting CPS”, as well as ‘most commonly’ is replaced with ‘a proven strategy’.

Line 30 – UDP-N-acetyl-d-galactosaminuronic acid; the d should be a small capital D. Check for other examples. Also, this is an activated donor substrate – not a monomer as described.

‘d’ has been changed to ‘D’. ‘a monomer’ has been replaced with ‘an activated donor substrate for Vi synthesis’.

Line 100 - is there data supporting the claim that TviE directly “controls” Vi length or is this an assumption? We replaced “controlling Vi length” with “controlling Vi polymerization” to represent previous reports.

Line 120 – Measured transcripts are not a convincing marker for expression. The level of stable protein is the critical issue. I would refer here to the later data with “see below” to indicate overall protein levels seem unchanged. Please see our responses above to this reviewer’s comments on the C127 mutant. We also note that there are no commercial antibodies available for TviD and TviE.

Line 316-318 – there seems to be something missing from this sentence. ‘than those from neutrophils incubated with WT, Vi negative, or hypo Vi capsule variants’ is added to the referred description (Line 322).

Line 410 – spelling - Klebsiella pneumoniaE
‘e’ is added.

REVIEWER COMMENTS

Reviewer #1 (Remarks to the Author):

The authors have made changes in the text that address most of my concerns. While it would have been nice to see a morphological effect in the hypercapsule mutants, for example via microscopy of india ink stained mutants, this reviewer acknowledges that it is unclear if it would work with Vi capsule. Given that the other phenotypic data on hypercapsule mutants is convincing and the manuscript is more carefully formulated I do not have any more concerns.

Reviewer #4 (Remarks to the Author):

As I stated with the original, I find this to be an interesting study but I still have concerns about unnecessary over interpretation. Some of my earlier concerns have been addressed but other issues remain. There are also instances throughout the text where the language is not as precise as it should be.

I still feel that separation of mutants into multiple categories is unnecessarily complicated without more detailed insight. Hyper- and hyper- should be sufficient.

Line 33: the text describes the polymer being acylated and has been amended with the addition of "which can be the first or last step...". The language here is imprecise. First, the added text implies that either option can exist, rather than it being one or the other, with the exact sequence not yet established. Also, strictly speaking, it isn't the polymer being acylated, it is the reducing terminal sugar, possibly before polymerization.

Line 86: repeats the point above but doesn't need to be stated again here since the same thing is also explained in the Fig 1 legend (it is helpful to have it in the legend).

Line 100: does TviE really control polymerization or just simply polymerize? In my view, the term 'control' invokes a different activity.

Line 117 and elsewhere, and Fig 1 legend: I commented on the earlier version that there is a need for caution in interpretations of polymer molecular weight - or length - based solely on SDS-PAGE migration. Unfortunately, the terminology is still present in this version with still no explanation of the limitations. SDS-PAGE migration is potentially influenced by chemistry (not just length) and intensity in western immunoblots is presumably affected by O-acetylation levels.

Line 138: I don't think mucoidity is a real word. I think the authors mean mucoidy.

Line 140: what "other bacteria"? Please give some examples.

Line 158: I still have concerns about the strength and narrow perspective of conclusions concerning the functional role of the "groove" in the TviE structure. A role in acceptor binding is certainly one possibility but it is not proven at this stage. In general, these kinds of binding surfaces are broad, so it can't be assumed (as the authors do) that a single residue within that site will influence acceptor binding (and by extension here, polymerization). The authors offer no precedent to support their argument and I think the interpretation is getting too far ahead of the data.

Line 161: I think the text here should be specific to acceptor binding rather than catalysis.

Line 166: what is an "in vivo molecular tool"?

Line 201: I think the assumption that C127 is required for O-acetyl transfer is an overstatement when based solely on proximity to a catalytic residue. There are many examples where some residues within a catalytic pocket can be changed with no effect on catalysis.

Line 654: I had asked about measures to standardize the western blot quantification and rule out over saturation. Given the importance assigned to this data, I was looking for something more systematic and compelling. The added text in the methods and rebuttal don't resolve the question.

Point-by-point response to reviewer #4

We thank this reviewer for his/her additional comments and suggestions on the manuscript. Our point-by-point responses are as follows. Furthermore, we have highlighted (in blue) the textual changes made in the updated version of the manuscript. Please keep in mind that we left the revised texts from the previous round highlighted in red for your convenience.

Reviewer #4 (Remarks to the Author):

As I stated with the original, I find this to be an interesting study but I still have concerns about unnecessary over interpretation. Some of my earlier concerns have been addressed but other issues remain. There are also instances throughout the text where the language is not as precise as it should be.

I still feel that separation of mutants into multiple categories is unnecessarily complicated without more detailed insight. Hyper- and hyper- should be sufficient.

Although a significant number of super/intermediate Vi capsule strains have hyper Vi capsule phenotypes, merging this intermediate group with the hyper Vi capsule category is not ideal. This is due to the observations that the intensity of super/intermediate Vi capsule strains is lower than that of hyper Vi capsule strains but higher than that of WT. Similarly, the phenotypes of super/intermediate Vi capsule strains differ from those of the WT group and hyper Vi capsule groups. Based on the reviewer's feedback, we've changed the term "super/intermediate Vi capsule" to "intermediate Vi capsule." We believe this is the best strategy to report this group with intermediate phenotypes.

Line 33: the text describes the polymer being acylated and has been amended with the addition of "which can be the first or last step...". The language here is imprecise. First, the added text implies that either option can exist, rather than it being one or the other, with the exact sequence not yet established. Also, strictly speaking, it isn't the polymer being acylated, it is the reducing terminal sugar, possibly before polymerization.

Clarified (Lines 33-34).

Line 86: repeats the point above but doesn't need to be stated again here since the same thing is also explained in the Fig 1 legend (it is helpful to have it in the legend).

Removed.

Line 100: does TviE really control polymerization or just simply polymerize? In my view, the term 'control' invokes a different activity.

Changed (Line 100).

Line 117 and elsewhere, and Fig 1 legend: I commented on the earlier version that there is a need for caution in interpretations of polymer molecular weight - or length - based solely on SDS-PAGE migration. Unfortunately, the terminology is still present in this version with still no explanation of the limitations. SDS-PAGE migration is potentially influenced by chemistry (not just length) and intensity in western immunoblots is presumably affected by O-acetylation levels.

The updated version of the manuscript discusses limitations (Lines 118-123).

Line 138: I don't think mucoidity is a real word. I think the authors mean mucoidy.

"Mucoidity" is replaced with "mucoidy" throughout the manuscript.

Line 140: what "other bacteria"? Please give some examples.

Two examples are added (Line 143).

Line 158: I still have concerns about the strength and narrow perspective of conclusions concerning the functional role of the "groove" in the TviE structure. A role in acceptor binding is certainly one possibility but it is not proven at this stage. In general, these kinds of binding surfaces are broad, so it can't be assumed (as the

authors do) that a single residue within that site will influence acceptor binding (and by extension here, polymerization). The authors offer no precedent to support their argument and I think the interpretation is getting too far ahead of the data.

We made changes to indicate that further research is necessary to formally test this potential mechanism and other alternative mechanisms (Lines 161-168).

Line 161: I think the text here should be specific to acceptor binding rather than catalysis. "catalysis" has been changed to "acceptor binding" (Line 164).

Line 166: what is an "in vivo molecular tool"?

Clarified. "*in vivo*" means "*S. Typhi*" (Line 171).

Line 201: I think the assumption that C127 is required for O-acetyl transfer is an overstatement when based solely on proximity to a catalytic residue. There are many examples where some residues within a catalytic pocket can be changed with no effect on catalysis.

We made changes (Lines 207-209).

Line 654: I had asked about measures to standardize the western blot quantification and rule out over saturation. Given the importance assigned to this data, I was looking for something more systematic and compelling. The added text in the methods and rebuttal don't resolve the question.

The updated version of the manuscript (Lines 656-661) describes additional relevant details.